# Dilated convolution neural operator for multiscale partial differential equations

## Abstract

This paper introduces a data-driven operator learning method for multiscale partial differential equations, with a particular emphasis on preserving high-frequency information. Drawing inspiration from the representation of multiscale parameterized solutions as a combination of low-rank global bases (such as low-frequency Fourier modes) and localized bases over coarse patches (analogous to dilated convolution), we propose the Dilated Convolutional Neural Operator (DCNO). The DCNO architecture effectively captures both high-frequency and low-frequency features while maintaining a low computational cost, through a combination of convolution and Fourier layers. We conduct experiments to evaluate the performance of DCNO on various datasets, including the multiscale elliptic equation, its inverse problem, Navier-Stokes equation, and Helmholtz equation. We show that DCNO strikes an optimal balance between accuracy and computational cost, and offers a promising solution for mulitscale operator learning.

## 1 Introduction

In recent years, operator learning methods such as Fourier neural operator (FNO) (Li et al., 2020b), Galerkin transformer (GT) (Cao, 2021), and deep operator network (DeepONet) (Lu et al., 2021) have emerged as powerful tools for computing parameter-to-solution maps of partial differential equations (PDEs). In this paper, we focus on multiscale PDEs that encompass multiple temporal/spatial scales. These multiscale PDE models are widely prevalent in physics, engineering, and other disciplines, playing a crucial role in addressing complex practical problems such as reservoir modeling, atmosphere and ocean circulation, and high-frequency scattering.

A well-known challenge with neural networks is their tendency to prioritize learning low-frequency components before high frequencies—a phenomenon referred to as "spectral bias" or "frequency principle" (Rahaman et al., 2019; Xu et al., 2019). This presents challenges when applying neural networks to functions characterized by multiscale or high-frequency properties, adapting neural network architectures (Cai & Xu, 2019; Wang et al., 2021) have been proposed to capture high-frequency details . In the context of operator learning, existing methods such as FNO and GT have shown spectral bias when applied to multiscale PDEs, as observed in Liu et al. (2023). To address this issue and recover high-frequency features, Liu et al. (2023) introduced an approach based on hierarchical attention and $H^1$ loss. However, despite providing improved accuracy, the method's high computational cost and implemational complexity to some extent counterbalances its strength.

In this paper, we present a novel method that strikes a balance among accuracy, computational cost, and the preservation of multiscale features. Our approach utilizes a carefully designed architecture that combines the strengths of dilated convolutions and Fourier layers. Dilated convolutions (Holschneider et al., 1990), also known as atrous convolutions, expand the kernel of a convolution layer in a convolutional neural network (CNN) by introducing gaps (holes) between the kernel elements. This technique allows for selectively skipping input values with specific step sizes, effectively covering a larger receptive field over the input feature map without introducing extra parameters or computational overhead. As a result, we can efficiently capture high-frequency local details. On the other hand, we leverage Fourier layers to capture the smooth global components of the data. DCNO achieves higher accuracy compared to existing models while maintaining lower computational costs by utilizing efficient implementations of both convolution and Fourier layers.

This makes our approach well-suited for applications that require the preservation of multiscale features.

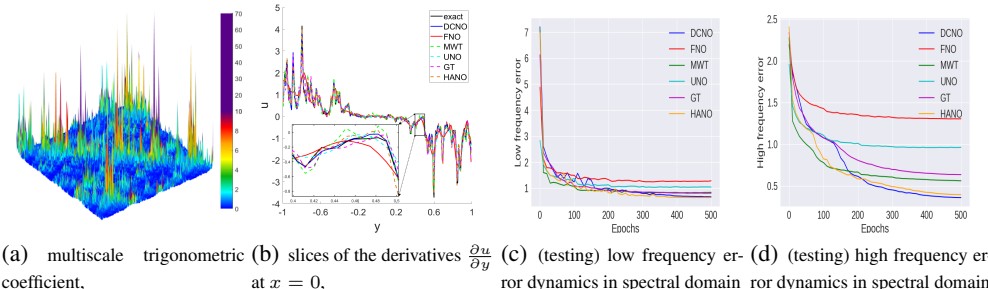

(a) multiscale trigonometric coefficient,

(b) slices of the derivatives $\frac{\partial u}{\partial y}$ at $x = 0$,

(c) (testing) low frequency error dynamics in spectral domain

(d) (testing) high frequency error dynamics in spectral domain

Figure 1.1: We demonstrate the effectiveness of the DCNO scheme using a challenging multiscale trigonometric benchmark. The coefficient and corresponding solution derivative are presented in (a) and (b), respectively (refer to Appendix A.1.2 for a detailed description, we note that all models are trained with $L^2$ loss). We observe that DCNO accurately captures the solution derivatives. In (c) and (d), we analyze the (testing) dynamics for high-frequency ($> 10\pi$) and low-frequency ($\leq 10\pi$) errors, respectively. It is evident that DCNO achieves the best performance in terms of both high-frequency and low-frequency errors (HANO is comparable but requires longer training time).

## 2 BACKGROUND AND RELATED WORK

### 2.1 MULTISCALE PDEs

We briefly introduce some representative multiscale PDEs in this section. One notable example is the class of multiscale elliptic PDEs, which involve coefficients varying rapidly and are often encountered in heterogeneous and random media applications, see details in Appendix A.1. For smooth coefficients, the coefficient to solution map can be effectively resolved by the FNO parameterization (Li et al., 2020b). However, when dealing with multiscale/rough coefficients, the presence of fast oscillation, high contrast ratios, and non-separable scales pose significant challenges from both scientific computing (Branets et al., 2009) and operator learning (Liu et al., 2023) perspectives. Other notable examples include the Navier-Stokes equation (see Appendix A.2), which models fluid flow and exhibits turbulence behavior at high Reynolds numbers, and the Helmholtz equation (see Appendix E), which models time-harmonic acoustic waves and is challenging to solve in the high wave number regime. In these multiscale PDEs, the accurate prediction of physical phenomena and properties necessitates the resolution of high-frequency components.

**Numerical Methods for Multiscale PDEs** Multiscale PDEs, even with fixed parameters, present a challenge for classical numerical methods, as their computational cost typically scales inversely proportional to the finest scale $\varepsilon$ of the problem. To overcome this issue, multiscale solvers have been developed by incorporating microscopic information to achieve computational cost independent of $\varepsilon$. One such technique is numerical homogenization (Engquist & Souganidis, 2008), which identifies low-dimensional approximation spaces adapted to the corresponding multiscale operator. Similarly, fast solvers like multilevel/multigrid methods (Hackbusch, 2013; Xu & Zikatanov, 2017) and wavelet-based multiresolution methods (Brewster & Beylkin, 1995; Beylkin & Coult, 1998) may face limitations when applied to multiscale PDEs (Branets et al., 2009), while multilevel methods based on numerical homogenization techniques, such as Gamblets (Owhadi, 2017), have emerged as a way to discover scalable multilevel algorithms and operator-adapted wavelets for multiscale PDEs. In recent years, there has been increasing exploration of neural network methods for solving multiscale PDEs despite the spectral bias or frequency principle (Rahaman et al., 2019; Ronen et al., 2019; Xu et al., 2019) indicating that deep neural networks (DNNs) often struggle to effectively capture high-frequency components of functions. Specifically designed neural solvers (Li et al., 2020a; Wang et al., 2021; Li et al., 2023) have been developed to mitigate the spectral bias and accurately solve multiscale PDEs (with fixed parameters).

Recent advancements in multiscale computational methods, as demonstrated in Målqvist & Peterseim (2014); Hauck & Peterseim (2023), have revealed that an exponentially decaying global basis achieves an optimal rate of convergence. Moreover, localizing the basis to coarse patches can help achieve the optimal trade-off between computational cost and accuracy. These findings serve as inspiration for our adoption of dilated convolution to extract local and high-frequency features. On the implementation level, there are two possible approaches. The first approach involves hierarchical decomposition, similar as multigrid or multilevel methods. These methods leverage a hierarchical structure to capture both global and local features effectively. The second approach is the global-local decomposition, as discussed by Benner et al. (2018), which combines long-range low-rank components with localized components. In our study, we explore the latter approach by employing an alternating architecture consisting of Fourier layers and convolutional layers.

## 2.2 NEURAL OPERATOR FOR MULTISCALE PDES

Neural operators, as proposed by Li et al. (2020b); Gupta et al. (2021), have shown great promise in capturing the input-output relationship of parametric partial differential equations (PDEs). However, multiscale PDEs introduce new challenges for neural operators. Fourier or wavelet transforms, which are central to the construction of Li et al. (2020b); Gupta et al. (2021), may not always be effective, even for multiscale PDEs with fixed parameters. Moreover, while universal approximation theorems exist for FNO-type models (Kovachki et al., 2021), achieving a meaningful convergence rate often requires "excessive smoothness" that may be absent for multiscale PDEs. Additionally, aliasing errors becomes significant in multiscale PDEs (Bartolucci et al., 2023), raising concerns about continuous-discrete equivalence. The work by Liu et al. (2023) addresses the issue of spectral bias in (multiscale) operator learning and highlights the challenges faced by existing neural operators in capturing high-frequency components of multiscale PDEs. These neural operators tend to prioritize the fitting of low-frequency components over high-frequency ones, limiting their ability to accurately capture fine details. To overcome this limitation, Liu et al. (2023) proposes a new architecture for multiscale operator learning that leverages hierarchical attention mechanisms and a tailored loss function. While these innovations help reduce the spectral bias and improve the prediction of multiscale solutions, it is worth noting that hierarchical attention induces a significant computational cost.

## 2.3 DILATED CONVOLUTION

In this paper, we focus on utilizing dilated convolutions to capture the high-resolution components of the data due to their simplicity and efficiency. Dilated convolution, also known as atrous convolution, was initially developed in the "algorithme à trous" for wavelet decomposition (Holschneider et al., 1990). Its primary purpose was to increase image resolution and enable dense feature extraction without additional computational cost in deep convolutional neural networks (CNNs) by inserting "holes" or zeros between pixels in convolutional kernels. By incorporating dilated convolution, networks can enlarge receptive fields, capture longer-range information, and gather contextual details, which are crucial for dense prediction tasks such as semantic segmentation. Various approaches have been proposed to leverage dilated convolution for this purpose (Yu & Koltun, 2015; Wang et al., 2018), and have demonstrated comparable results compared with U-Net and attention based models.

More recently, dilated convolution has also found applications in operator learning, such as the Dil-ResNet used for simulating turbulent flow (Stachenfeld et al., 2022). However, our work demonstrates that using dilated convolution alone is not sufficient to accurately capture the solution. Instead, we propose an interwoven global-local architecture of Fourier layers with dilated convolution layers. Furthermore, while Dil-ResNet requires up to 10 million training steps to achieve satisfactory results, our model offers a more efficient approach. It is worth noting that there are many alternative approaches to extract multiscale features, inspired by developments in numerical analysis and computer vision. These include hierarchical matrix methods (Fan et al., 2019), hierarchical attention (Liu et al., 2021; 2023), U-Net (Ronneberger et al., 2015) and U-NO (Rahman et al., 2022), wavelet-based methods (Gupta et al., 2021), among others. In Section 4, we will conduct a comprehensive benchmark of these different multiscale feature extraction techniques to evaluate their performance.

While dilated convolutions can expand the receptive field, our understanding from multiscale computational methods suggests that they primarily provide accuracy on a coarse scale. To achieve higher accuracy, it is essential to accurately extract both local and global features. This is precisely why we integrate Fourier layers with (dilated) convolution layers, thereby enhancing overall accuracy.

## 3 METHODS

We adopt a data-driven approach to approximate the operator $\mathcal{S} : \mathcal{H}_1 \mapsto \mathcal{H}_2$ as in references Li et al. (2020b); Cao (2021); Liu et al. (2023). The operator $\mathcal{S}$ maps between two infinite-dimensional Banach spaces $\mathcal{H}_1$ and $\mathcal{H}_2$, and aims to find the solution to the parametric partial differential equation (PDE) $\mathcal{L}_{\boldsymbol{a}}(\boldsymbol{u}) = f$, where the input/parameter $\boldsymbol{a} \in \mathcal{H}_1$ is drawn from a distribution $\mu$, and the corresponding output/solution $\boldsymbol{u} \in \mathcal{H}_2$.

To be specific, in this paper, our objective is to address the following operator learning problems:

- Approximating the nonlinear mapping $\mathcal{S} : \boldsymbol{a} \mapsto \boldsymbol{u} := \mathcal{S}(\boldsymbol{a})$ from the varying parameter $\boldsymbol{a}$ to the solution $\boldsymbol{u}$.

- Solving the inverse coefficient identification problem, which involves recovering the coefficient from a noisy measurement $\hat{\boldsymbol{u}}$ of the solution $\boldsymbol{u}$. In this scenario, we aim to approximate $\mathcal{S}^{-1} : \hat{\boldsymbol{u}} \mapsto \boldsymbol{a} := \mathcal{S}^{-1}(\hat{\boldsymbol{u}})$.

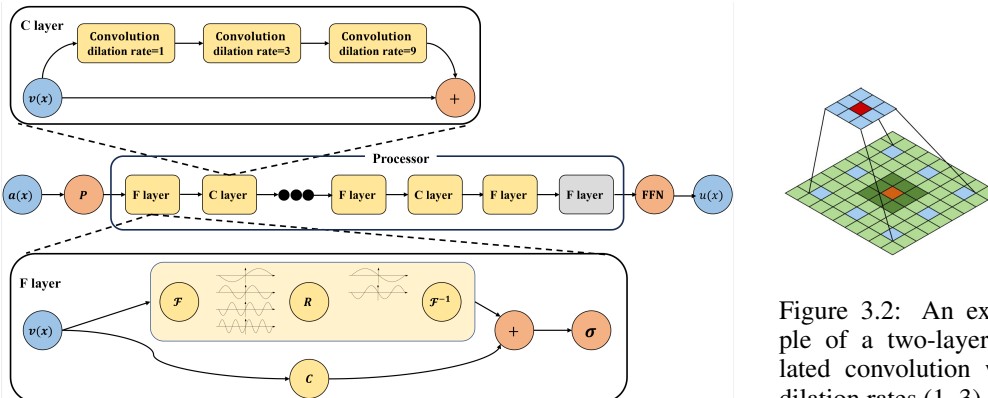

Figure 3.1: The architecture of the DCNO neural operator.

Figure 3.2: An example of a two-layer dilated convolution with dilation rates (1, 3).

### 3.1 MODEL ARCHITECTURES

In our model, we employ an Encode-Process-Decode architecture (Sanchez-Gonzalez et al., 2018; 2020), as shown in Figure 3.1.

- The encoder incorporates a patch embedding function denoted as P, which utilizes a convolutional neural network (CNN) that is described in detail in Appendix B.3.1. This step is performed to lift the input $a(x)$ to a higher-dimensional channel (feature) space.

- The processor part of the model comprises alternating Fourier layers (F layers) and Convolution layers (C layers). The role of the F layer is to approximate the low-frequency components, while the C layer is responsible for extracting high-frequency features. This alternating approach allows the DCNO model to effectively handle both low-frequency and high-frequency components present in the data. To gain further insights into the influence of the F and C layers, an ablation study is conducted, as described in Appendix B.3.1. This study provides information about the impact of these layers on the model's performance and enhances our understanding of their significance within the overall architecture.

- The decoder in our model adopts a three-layer feedforward neural network ($FFN$).

**F layers:**    The Fourier layers, as proposed in Li et al. (2020b), consist of two main components that operate on the input $v(x)$. In the first component, the input undergoes the Fourier transform, followed by a linear transform $R$ acting only on the lower Fourier modes while filtering out the higher modes. The modified input is then obtained by applying the inverse Fourier transform. The first component of the Fourier layer aims to preserve low-frequency global information while reducing the influence of high-frequency components. The second component of the Fourier layers incorporates a convolutional neural network (CNN) with a kernel size of 3, replacing the local (pointwise) linear transform $W$ used in Li et al. (2020b). This choice is motivated by the findings of Liu et al. (2023), which suggest that a CNN with a small kernel size may help extract high-frequency information more effectively compared to the pointwise linear transformation used in the original FNO implementation. Additionally, the outputs from both components are combined using the GELU activation function.

It is important to note that while the second component of the F layer helps capture some high-frequency details, relying solely on this part is not sufficient to accurately capture high-frequency information. This limitation is why the Fourier neural operator ($FNO$) approach may not perform well for multiscale PDE problems. We conduct a detailed ablation study in Appendix B.3.2 to further investigate this issue.

**C layers:**    Each convolution layer includes three convolutional neural networks with increasing dilation rates, each utilizing a kernel of size 3 and followed by a GELU activation function. These convolutional neural networks employ dilation rates of $(1, 3, 9)$. The dilation rate determines the spacing between the points with which each point is convolved. A dilation rate of 1 corresponds to a regular convolution where each point is convolved with its immediate neighbors. Larger dilation rates, such as 3 and 9, expand the receptive field of each point to include more distant points. Figure 3.2 illustrates an example of a two-layer dilated convolution with dilation rates of $(1, 3)$. In the first layer, each point is convolved with its neighbors at a distance of 1, and in the second layer, each point is convolved with its neighbors at a distance of 3. As a result, the central red cell has a $9 \times 9$ receptive field. By incorporating multiple dilation rates, the model can capture long-range dependencies and maintain communication between distant points. This approach enhances the model's ability to capture both local and global information. Residual connections are applied to alleviate the vanishing gradient problem.

In summary, the combination of convolution layers with multiple dilation rates effectively enlarges the receptive fields of the network, facilitating the aggregation of global information and leveraging the advantages of convolutions in extracting localized features (see Section 4.3). However, in most cases in operator learning, relying solely on convolutional layers may not yield satisfactory results, and the combination with Fourier layers can boost performance. See the ablation results in Appendix B.3.1 for more details.

## 3.2    WEIGHTED LOSS FUNCTION

Loss functions play a crucial role in effectively training neural network models. The conventional $L^2$ loss, denoted as $\mathcal{L}^{L^2}(\boldsymbol{v}, \boldsymbol{u}) = \sqrt{\sum_{j=1}^{N} |\boldsymbol{v}_j - \boldsymbol{u}_j|^2}$, can be equivalently expressed in the Fourier domain as $\mathcal{L}^{L^2}(\boldsymbol{v}, \boldsymbol{u}) = \sqrt{\sum_{\xi=-N/2+1}^{N/2} |\hat{\boldsymbol{v}}_\xi - \hat{\boldsymbol{u}}_\xi|^2}$. $\hat{\boldsymbol{v}}$ and $\hat{\boldsymbol{u}}$ are the Fourier transforms of $\boldsymbol{v}$ and $\boldsymbol{u}$, respectively. For multiscale problems, it is natural to consider using the $H^1$ loss function, which incorporates derivatives in addition to the $L^2$ loss. In the Fourier domain, the $H^1$ loss can be defined as $\mathcal{L}^{H^1}(\boldsymbol{v}, \boldsymbol{u}) := \sqrt{\sum_{\xi=-N/2+1}^{N/2} (1 + 4\pi^2|\xi|^2)|\hat{\boldsymbol{v}}_\xi - \hat{\boldsymbol{u}}_\xi|^2}$. This loss function introduces a term proportional to $|\xi|^2$ in the summation, where $|\xi|$ represents the frequency. By including this term, the $H^1$ loss emphasizes high-frequency components. In practice, the frequency distribution of the solution is often unknown beforehand. Hence, inspired by the work of Liu et al. (2023), we adopt a weighted loss function denoted as $\mathcal{L}^T(\boldsymbol{v}, \boldsymbol{u}) := \sqrt{\sum_{|\xi|=1}^{T} 4\pi^2|\xi|^2|\hat{\boldsymbol{v}}_\xi - \hat{\boldsymbol{u}}_\xi|^2 + \sum_{\xi=-N/2+1}^{N/2} |\hat{\boldsymbol{v}}_\xi - \hat{\boldsymbol{u}}_\xi|^2}$, where weights are only applied to the first $T$ frequencies. This approach aims to strike a balance between preserving low-frequency information and reducing high-frequency errors. For more details, please refer to Appendix D.

## 4 EXPERIMENTS

In this section, we present numerical experiments comparing DCNO with different operator models based on several metrics, including relative $L^2$ error, parameter count, memory consumption, and training time per epoch. The goal of these experiments is to assess the performance of various operator models through different tasks, which include multiscale elliptic equations, time-dependent Navier-Stokes equations, and inverse coefficient identification for multiscale elliptic equations. Additionally, we explore the Helmholtz equation in Appendix E and a Navier-Stokes example in a different setup (de Hoop et al., 2022) in Appendix F. The results consistently demonstrate that the DCNO model outperforms other operator models. It achieves superior accuracy, robustness, and cost-accuary trade-off in all the considered scenarios.

The experiments in the study are trained using either relative $L^2$ loss or weighted $L^T$ loss unless stated otherwise, and denoted by $(L^2)$ or $(L^T)$, respectively. For more detailed information on the experimental setup, including specific configurations and parameters, please refer to Appendix B.1.

**Benchmark models:** We compare the DCNO model with the following recent successful operators: **Fourier Neural Operator (FNO)**, a neural operator method based on the Fourier transform (Li et al., 2020b); **Multiwavelet-based Operator (MWT)** (Gupta et al., 2021), a neural operator based on multiwavelet transform; **U-shaped Neural Operator (U-NO)**, (Rahman et al., 2022) a neural operator combining FNO and U-Net (Ronneberger et al., 2015) architectures and is considered as a superior alternative to U-NET; **Galerkin Transformer (GT)**, (Cao, 2021) a neural operator utilizing an encoder which rearranges the order of multiplication in vanilla attention for feature extraction; **Hierachical Attention Neural Operator (HANO)**, (Liu et al., 2023) a hierarchical attention neural operator inspired by the hierarchical matrix approach; **Dilated ResNet (Dil-ResNet)**, (Stachenfeld et al., 2022) a method combining the encode-process-decode paradigm with dilated convolution.

### 4.1 MULTISCALE ELLIPTIC EQUATIONS

We examine the effectiveness of the DCNO model on the multiscale elliptic equation, given by the following second-order linear elliptic equation,

$$\begin{cases} -\nabla \cdot (a(x)\nabla u(x)) = f(x), & x \in D \\ \qquad\qquad u(x) = 0, & x \in \partial D \end{cases} \tag{4.1}$$

with rough coefficients and Dirichlet boundary conditions. Our goal is to approximate the operator $\mathcal{S} : L^\infty(D; \mathbb{R}_+) \to H_0^1(D; \mathbb{R})$, which maps the coefficient function $a(x)$ to the corresponding solution $u$. We assess the model on two-phase Darcy rough coefficients (Darcy rough) given in Liu et al. (2023) where the coefficients are significantly rougher compared to the well-known benchmark proposed in Li et al. (2020b). We also consider multiscale trigonometric coefficients with higher contrast, following the setup in Owhadi (2017); Liu et al. (2023). The coefficients and solutions are displayed in Figure 4.1. Further details on the data generation and can be found in Appendix A.1.

The experimental results for multiscale elliptic equations are presented in Table 1. The models are trained with either $L^2$ or weighted $L^T$ loss, and evaluated by both $L^2$ and frequency based $L^T$ error metrics. The results can be summarized below,

- DCNO achieves the lowest relative error compared to other neural operators at various resolutions, and the errors remain approximately invariant with the resolution. Compared to FNO, DCNO has a remarkable accuracy improvement of 73% and 64% in the cases of Darcy rough and multiscale trigonometric, respectively, while maintaining the second fewest parameters and only requireing 39% more training time compared to FNO. Furthermore, we observe that DCNO achieves the best cost-accuracy trade-off among all the neural operators we tested in Figure 4.2.

- Attention-based models GT and HANO suffer from high computational costs in terms of training time and memory. Although HANO achieves the second-best accuracy, DCNO outperforms HANO by a significant margin while requiring fewer computational resources.

- Dil-ResNet, with its convolutional architecture, has the fewest parameters among all models. However, it requires more memory and longer training time per epoch compared to DCNO and FNO. Furthermore, its accuracy is not as ideal as the other models.

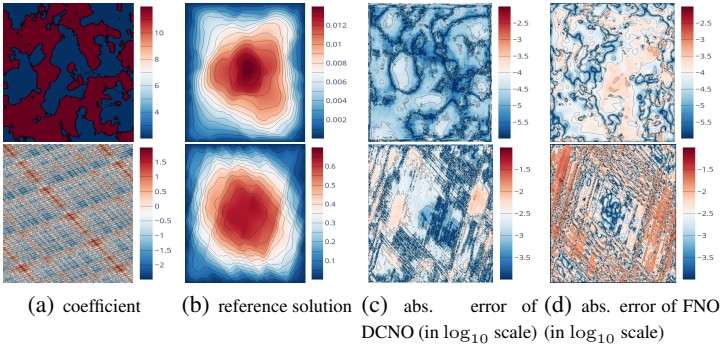

Figure 4.1: **Top:** Darcy rough example, (a) coefficient, (b) reference solution, (c) DCNO, absolute (abs.) error, (d) FNO, abs. error; **Bottom:** multiscale trignometric example, (a) coefficient (in $\log_{10}$ scale), (b) reference solution, (c) DCNO, abs. error (in $\log_{10}$ scale), (d) FNO, abs. error (in $\log_{10}$ scale).

Figure 4.2: Cost-accuracy (in $\log_2$ scale) trade-off

In addition, the reduction of spectral bias for DCNO and a comparison with other models can be found in Appendix C. These results highlight the superior performance of the DCNO model in terms of accuracy and efficiency, making it a favorable choice for solving multiscale elliptic equations. Also, we observe from Table 1 that even trained with $L^2$ loss, DCNO outperforms most models trained with weighted $L^T$ loss. Only HANO achieves a comparable level of accuracy as DCNO, but it requires more than 3 times the training time [1]

| Model | Parameters $\times 10^6$ | Memory (GB) | Time per epoch(s) | Darcy rough $L^2$ | $L^T$ | Trigonometric $L^2$ | $L^T$ |
|---|---|---|---|---|---|---|---|
| FNO($L^2$) | 2.37 | 1.79 | 5.71 | 1.749 | 15.192 | 1.803 | 30.264 |
| FNO($L^T$) | 2.37 | 1.79 | **5.65** | 1.643 | 13.671 | 1.760 | 15.191 |
| MWT($L^2$) | 9.81 | 2.54 | 18.14 | 1.301 | 5.437 | 0.988 | 11.158 |
| MWT($L^T$) | 9.81 | 2.54 | 17.94 | 1.225 | 4.621 | 0.870 | 6.056 |
| U-NO($L^2$) | 16.39 | **1.57** | 9.82 | 1.324 | 9.727 | 1.370 | 19.958 |
| U-NO($L^T$) | 16.39 | **1.57** | 9.84 | 1.104 | 7.915 | 1.184 | 7.910 |
| GT($L^2$) | 2.22 | 9.32 | 35.22 | 2.166 | 10.686 | 1.013 | 13.193 |
| GT($L^T$) | 2.22 | 9.32 | 35.25 | 1.739 | 5.805 | 0.988 | 7.860 |
| HANO($L^2$) | 13.37 | 9.87 | 27.42 | 1.119 | 5.158 | 0.743 | 7.866 |
| HANO($L^T$) | 13.37 | 9.87 | 27.42 | 0.674 | 2.507 | 0.645 | 4.368 |
| DIL-RESNET($L^2$) | **0.58** | 5.71 | 10.67 | 7.110 | 24.462 | 2.301 | 28.513 |
| DIL-RESNET($L^T$) | **0.58** | 5.71 | 10.69 | 5.202 | 11.620 | 1.848 | 7.512 |
| DCNO($L^2$) | 1.74 | 2.68 | 7.84 | 0.673 | 4.237 | 0.749 | 7.791 |
| DCNO($L^T$) | 1.74 | 2.68 | 7.88 | **0.446** | **1.802** | **0.631** | **3.689** |

Table 1: Benchmarks on multiscale elliptic equations. Performance are measured with relative $L^2$ errors ($\times 10^{-2}$), number of parameters, memory consumption for a batch size of 8, and time per epoch for resolution $s = 256$ of the Darcy rough example during the training process, the notation ($L^2$) or ($L^T$) indicates whether the model was trained using the $L^2$ loss or the weighted loss, respectively.

## 4.2 NAVIER-STOKES EQUATION

In this section, we focus on the 2D Navier-Stokes equation in vorticity form on the unit torus $\mathsf{T}$, as benchmarked in (Li et al., 2020b)(see Appendix A.2 for details). The vorticity variable is denoted as $\omega(x, t)$, where $x \in \mathsf{T}$ represents the spatial domain and $t \in [0, T]$ represents the time interval. The goal is to learn the operator $\mathcal{S} : w(\cdot, 0 \le t < T_0) \to w(\cdot, T_0)$, which maps the vorticity from time 0 to $T_0 - 1$ to the vorticity at time $T_0$. This operator can be applied recursively until reaching

---

[1]The current version of HANO is based on SWIN (Liu et al., 2021) implementation.

the final time $T$, with the "rollout" strategy used in Li et al. (2020b) and Rahman et al. (2022). .
In our experiments, we consider different viscosities $\nu \in \{1e-3, 1e-4, 1e-5, 1e-6\}$, with
the final time $T$ adjusted accordingly as the flow becomes more chaotic with decreasing viscosities.
This strategy involves predicting the vorticity at each time step using a recurrence relation: $\tilde{w}_t = \mathcal{G}(\tilde{w}_{t-1}, \tilde{w}_{t-2}, \cdots, \tilde{w}_{t-T_0})$ and $\tilde{w}_i = w_i$ if $0 \leq t < T_0$, where $\tilde{w}_i$ is the predicted vorticity and $w_i$
is the true vorticity. The operator $\mathcal{G}$ is approximated by various neural operators. By evaluating the
performance of the DCNO model and other benchmark operators, our aim is to assess their ability
to accurately predict the complex dynamics of fluid flow over time.

Table 2 presents a comprehensive summary of the Navier-Stokes experiment results. For lower
Reynolds numbers, FNO, MWT, U-NO, HANO and DCNO achieve similar levels of accuracy.
However, as the Reynolds number increases, the DCNO models demonstrate a notable advantage in
accuracy. This highlights the effectiveness of the DCNO models in capturing the complex dynamics
of fluid flow and accurately predicting vorticity, especially in scenarios with higher Reynolds num-
bers. Overall, the results demonstrate that DCNO consistently provides more accurate predictions
compared to other methods while maintaining a reasonable computational cost.

| Model | Parameters $\times 10^6$ | Memory Requirement (GB) | Time per epoch (s) | $\nu = 1e-3$ $T_0 = 10s$ $T = 50s$ | $\nu = 1e-4$ $T_0 = 10s$ $T = 25s$ | $\nu = 1e-5$ $T_0 = 10s$ $T = 20s$ | $\nu = 1e-6$ $T_0 = 6s$ $T = 15s$ |
|---|---|---|---|---|---|---|---|
| FNO | 2.37 | 0.23 | **23.13** | 0.406 | 4.561 | 7.820 | 5.280 |
| MWT | 9.81 | 0.32 | 77.11 | 0.388 | 4.103 | 8.424 | 4.957 |
| U-NO | 11.91 | **0.22** | 96.02 | 0.454 | **3.573** | 6.923 | 4.588 |
| GT | 2.23 | 1.117 | 122.72 | 2.516 | 8.352 | 11.253 | 7.149 |
| HANO | 3.27 | 0.49 | 40.70 | 0.375 | 4.405 | 7.109 | 4.101 |
| DIL-RESNET | **0.59** | 0.72 | 43.55 | 1.100 | 9.833 | 14.870 | 9.810 |
| DCNO | 3.06 | 0.38 | 31.68 | **0.348** | 4.209 | **6.239** | **3.227** |

Table 2: Benchmarks on Navier Stokes equation. Performance are measured with relative $L^2$ errors
$(\times 10^{-2})$, number of parameters, memory consumption for a batch size of 16, and time per epoch
for $(\nu = 1e-5, T_0 = 10s, T = 20s)$ during the training process.

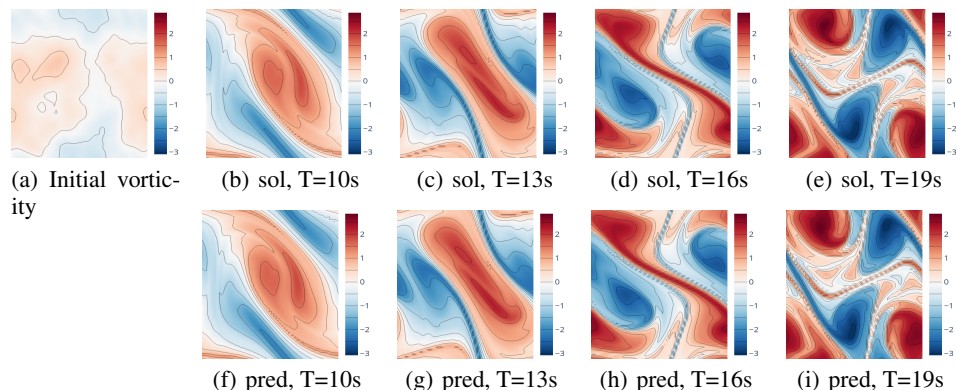

| (a) Initial vortic­ity | (b) sol, T=10s | (c) sol, T=13s | (d) sol, T=16s | (e) sol, T=19s |

| (f) pred, T=10s | (g) pred, T=13s | (h) pred, T=16s | (i) pred, T=19s |

Figure 4.3: The vorticity field generated by DCNO method for the Navier-Stokes equation with
viscosity $10^{-5}$, sol stands for solution, pred stands for predictions.

## 4.3 INVERSE COEFFICIENT IDENTIFICATION FOR MULTISCALE ELLIPTIC PDES

In this section, we address an inverse coefficient identification problem using the same data as the
previous example of multiscale elliptic PDE in section 4.1. Inverse problems play a crucial role
in various scientific fields, including geological sciences and medical imaging. However, these
problems often exhibit poor stability compared to their corresponding forward problems (see B.4),
even with advanced regularization techniques (Kirsch et al., 2011; Gottschling et al., 2020; Scarlett
et al., 2022). In this example, our objective is to learn an approximation to an ill-posed operator
$\mathcal{S}^{-1} : H_0^1(D) \mapsto L^\infty(D)$, where $\hat{u} = u + \epsilon N(u) \mapsto a$. Here, $\epsilon$ represents the level of Gaussian
noise added to both the training and evaluation data. The noise term $N(u)$ accounts for the sampling
distribution and data-related noise. This task is challenging due to the ill-posed nature of the problem
and the presence of noise.

The results of the inverse coefficient identification problem with noise are presented in Table 3. It is worth noting that the memory consumption and training time per epoch remain the same as reported in Table 1. The DCNO model outperforms other methods in this example, which highlights DCNO's ability to effectively address the challenges posed by this ill-posed inverse problem with noisy data. Interestingly, FNO and U-NO, known for their effectiveness in smoothing and filtering high-frequency modes, encounter difficulties in recovering targets that exhibit high-frequency characteristics, such as irregular interfaces, highly oscillatory coefficients, and the presence of Gaussian noise. In contrast, Dil-ResNet (and MWT to a lesser extent) performs significantly better in this specific problem, particularly when the noise level is higher. The use of dilated convolutions in Dil-ResNet proves advantageous in capturing high-frequency features. While HANO achieves the second-best overall accuracy, it comes at a considerable computational cost.

In Figure 4.4, we display the solution and predicted coefficients for the inverse coefficient identification problem at various levels of noise. Notably, even with a noise level of 10%, the predicted coefficient successfully recovers the interface present in the ground truth. This resilience to noise highlights the robustness and effectiveness of the coefficient prediction in capturing the underlying structure accurately.

| Model | Darcy rough | | | Trigonometric | | |
|---|---|---|---|---|---|---|
| | $\epsilon=0$ | $\epsilon=0.01$ | $\epsilon=0.1$ | $\epsilon=0$ | $\epsilon=0.01$ | $\epsilon=0.1$ |
| FNO | 28.154 | 28.267 | 29.725 | 56.044 | 56.043 | 56.239 |
| MWT | 9.361 | 12.571 | 20.818 | 11.042 | 12.679 | 18.529 |
| U-NO | 23.272 | 23.114 | 25.741 | 52.019 | 51.856 | 51.998 |
| GT | 12.021 | 14.539 | 23.145 | 25.489 | 27.186 | 41.729 |
| HANO | **2.502** | 9.400 | 19.204 | 8.859 | 10.637 | 17.479 |
| DIL-RESNET | 6.469 | 11.581 | 20.608 | 10.549 | 13.262 | 17.406 |
| DCNO | 2.737 | **8.765** | **17.042** | **7.723** | **8.110** | **9.512** |

Table 3: Relative error $(\times 10^{-2})$ of the inverse coefficient identification. Also see Figure 4.4 for solutions and predicted coefficients at various noise levels.

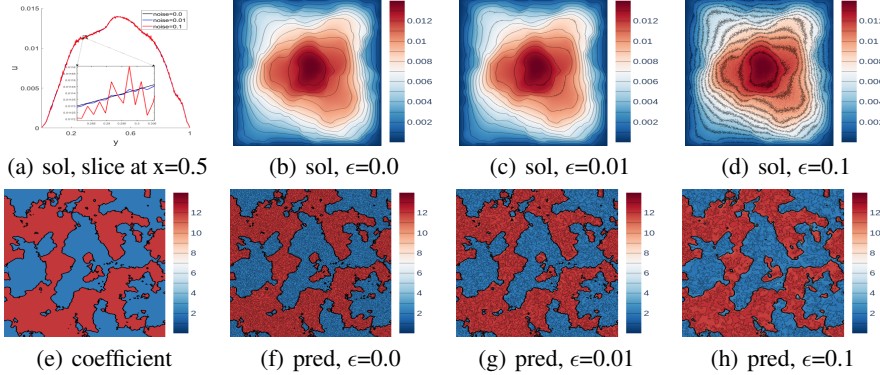

(a) sol, slice at x=0.5    (b) sol, $\epsilon=0.0$    (c) sol, $\epsilon=0.01$    (d) sol, $\epsilon=0.1$

(e) coefficient    (f) pred, $\epsilon=0.0$    (g) pred, $\epsilon=0.01$    (h) pred, $\epsilon=0.1$

Figure 4.4: DCNO inverse coefficient identification, sol stands for solution, pred stands for predicted coefficients.

## 5 CONCLUSION

In this paper, we introduce DCNO (Dilated Convolution Neural Operator) as a novel and effective method for learning operators in multiscale PDEs. DCNO combines the strengths of Fourier layers, which excel at representing low-frequency global components, with convolution layers that employ multiple dilation rates to capture high-resolution local details. This hybrid architecture empowers DCNO to surpass existing operator methods, offering a highly accurate and computationally efficient approach for learning operators in multiscale settings. Through extensive experiments, we demonstrate the effectiveness of DCNO in addressing multiscale PDEs, showcasing its superior performance and potential for a wide range of applications.

## REPRODUCIBILITY STATEMENT

We put the code for and also a link for datasets at the anonymous Github page `https://github.com/cesare4444/DCNO-ICLR2024`. Supplementary descriptions of the code are also provided in the page. The datasets for the Darcy rough (two-phase coefficients) example in Section 4.1 and the Navier-Stokes example in Section 4.2 are generated using the code from `https://github.com/zongyi-li/fourier_neural_operator`. We implemented $\mathcal{P}_1$ finite element method in MATLAB to solve the multiscale trigonometric example in Section 4.1, and in FreeFEM++ (Hecht, 2012) to solve the Helmholtz equation in Appendix E. We have included introductions of the relevant mathematical and data generation concepts in the Appendix.

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

## A  DATA GENERATION

### A.1  MULTISCALE ELLIPTIC PDES

Multiscale elliptic equations are a fundamental class of problems, exemplified by the following second-order elliptic equation in divergence form:

$$\begin{cases} -\nabla \cdot (a(x)\nabla u(x)) = f(x), & x \in D \\ \qquad\qquad\qquad u(x) = 0, & x \in \partial D \end{cases}$$

Here, the coefficient $a(x)$ satisfies $0 < a_{\min} \le a(x) \le a_{\max}$ for all $x \in D$, and $f \in H^{-1}(D;\mathbb{R})$ represents the forcing term. The coefficient-to-solution map is denoted as $\mathcal{S} : L^{\infty}(D;\mathbb{R}^+) \to H_0^1(D;\mathbb{R})$, where $u = \mathcal{S}(a)$. The coefficient $a(x)$ may exhibit rapid oscillations (e.g., $a(x) = a(x/\varepsilon)$ with $\varepsilon \ll 1$), high contrast ratios with $a_{\max}/a_{\min} \gg 1$, and even a continuum of non-separable scales. Handling rough coefficients poses significant challenges from both scientific computing (Branets et al., 2009) and operator learning perspectives. In following, we give the details of the two examples of multiscale elliptic equations benchmarked in this paper.

#### A.1.1  DARCY ROUGH EXAMPLE

Darcy flow, originally introduced by Darcy (Darcy, 1856), has diverse applications in modeling subsurface flow pressure, linearly elastic material deformation, and electric potential in conductive materials. The two-phase coefficients and solutions are generated using the approach outlined in https://github.com/zongyi-li/fourier_neural_operator/tree/master/data_generation. Given the computational domain $[0,1]^2$, the coefficients $a(x)$ are generated according to $a \sim \mu := \psi_\# \mathcal{N}\left(0, (-\Delta + cI)^{-2}\right)$, where $\Delta$ represents the Laplacian with zero Neumann boundary condition. The mapping $\psi : \mathbb{R} \to \mathbb{R}$ takes the value 12 for the positive part of the real line and 2 for the negative part, with a contrast of 6. The push-forward is defined in a pointwise manner. These generated datasets serve as benchmark examples for operator learning in various studies, including Li et al. (2020b), Gupta et al. (2021), and Cao (2021). The parameter $c$ can be used to control the "roughness" of the coefficient and corresponding solution. In the aforementioned references, the parameter $c$ is set as $c = 9$, while in Liu et al. (2023), a value of $c = 20$ is used to generate a rougher coefficient. The forcing term is fixed as $f(x) \equiv 1$. Solutions $u$ are obtained using a second-order finite difference scheme on a $512 \times 512$ grid. Lower-resolution datasets are created by sub-sampling from the original dataset through linear interpolation.

#### A.1.2  MULTISCALE TRIGONOMETRIC EXAMPLE

Multiscale trigonometric coefficients are described in Owhadi (2017), as an example of highly oscillatory coefficients. Given the domain $D$ $[-1,1]^2$, the coefficient $a(x)$ is specified as follows: $a(x) = \prod_{k=1}^{6} \left(1 + \frac{1}{2}\cos(a_k\pi(x_1+x_2))\right)\left(1 + \frac{1}{2}\sin(a_k\pi(x_2-3x_1))\right)$ Here, $a_k$ is uniformly distributed in $[2^{k-1}, 1.5 \times 2^{k-1}]$. The forcing term is fixed at $f(x) \equiv 1$. To obtain the reference solutions, the $\mathcal{P}_1$ Finite Element Method (FEM) is employed on a $1023 \times 1023$ grid. Lower-resolution datasets are generated by downsampling the higher-resolution dataset through linear interpolation.

We present the multiscale trigonometric coefficient, reference solution, and a comparison with other operator learning models in Figure A.1. Among the models considered, DCNO demonstrates superior accuracy in predicting function values and, more importantly, accurately captures the fine-scale oscillations. This is evident in the predicted derivatives shown in (c) of Figure A.1.

### A.2  NAVIER-STOKES EQUATION

The behavior of fluid flow on the unit torus is described by the Navier-Stokes equation in vorticity form. This equation is given by:

$$\begin{cases} \partial_t w(x,t) + u(x,t) \cdot \nabla w(x,t) = \nu\Delta w(x,t) + f(x), & x \in (0,1)^2, t \in (0,T] \\ \qquad\qquad\qquad \nabla \cdot u(x,t) = 0, & x \in (0,1)^2, t \in [0,T] \\ \qquad\qquad\qquad w(x,0) = w_0(x), & x \in (0,1)^2 \end{cases} \tag{A.1}$$

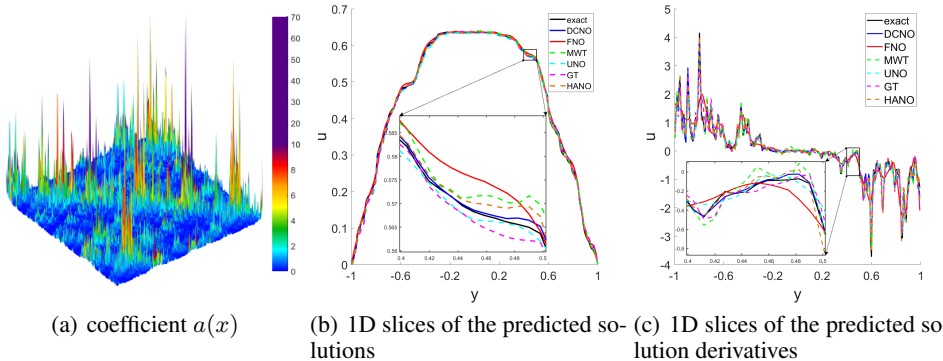

(a) coefficient $a(x)$     (b) 1D slices of the predicted solutions     (c) 1D slices of the predicted solution derivatives

Figure A.1: (a) multiscale trigonometric coefficient, (b)comparison of predicted solutions on the slice $x = 0$, (c) comparison of predicted derivative $\frac{\partial u}{\partial y}$ on the slice $x = 0$.

In the context of fluid dynamics, the variables used in the equations have the following interpretations:

- The velocity field is represented by the symbol $u$.

- The vorticity field is denoted as $w$, and it is defined as the curl of the velocity field, i.e., $w = \nabla \times u$.

- The initial vorticity distribution is denoted by $w_0$.

- The viscosity coefficient is represented by $\nu$.

- The forcing term is given by $f(x) = 0.1 \left( \sin \left( 2\pi (x_1 + x_2) \right) + \cos \left( 2\pi (x_1 + x_2) \right) \right)$.

The Reynolds number, denoted as Re, is a dimensionless parameter defined as $\mathrm{Re} := \frac{\rho u L}{\nu}$, where $\rho$ is the density (assumed to be 1), $u$ is the fluid velocity, and $L$ is the characteristic length scale of the fluid (set to 1). The Reynolds number is inversely proportional to the viscosity coefficient $\nu$. An increase in the Reynolds number tends to promote the transition of the flow to turbulence. The initial vorticity distribution $w_0(x)$ is generated from a probability measure $\mu$. Specifically, $w_0 \sim \mu$, where $\mu = \mathcal{N}\left(0, 7^{3/2}(-\Delta + 49I)^{-2.5}\right)$, and periodic boundary conditions are applied. In the data generation process, all data is produced on a $256 \times 256$ grid and then downsampled to a resolution of $64 \times 64$. The approach and code utilized for data generation can be found at the following URL: https://github.com/zongyi-li/fourier_neural_operator/tree/master/data_generation.

## B   SUPPLEMENTAL DETAILS OF THE EXPERIMENTS

### B.1   TRAINING AND EVALUATION SETUP

Unless stated otherwise, the train-val-test split datasets used consist of 1000, 100, and 100 samples, respectively, with a maximum of 500 training epochs and batch size of 8. The Adam optimizer is utilized with a decay of $1e - 4$ and a 1cycle learning rate scheduler (Smith & Topin, 2019). For the Navier-Stokes equations, the train-val-test split dataset has 5000, 500, and 500 samples, respectively, at a resolution of $64 \times 64$, again with a maximum of 500 epochs and batch size of 16. All experiments are executed on an NVIDIA A100 GPU.

### B.2   HYPERPARAMETER STUDY

We conducted a hyperparameter study to investigate the influence of different dilation rates in the C layers of the DCNO model for multiscale elliptic PDEs. The results of this study are summarized in Table 4. The dilation rates determine the configuration of the C layers in the DCNO model, where dilation rates of $(1, 3, 9)$ means that the C layers consist of three dilated convolutions with dilation

| Dilation | Time per epoch(s) | Darcy rough | | Trigonometric | |
|---|---|---|---|---|---|
| | | $L^2(\times 10^{-2})$ | $L^T(\times 10^{-2})$ | $L^2(\times 10^{-2})$ | $L^T(\times 10^{-2})$ |
| $(1, 3, 9)$ | 7.88 | 0.446 | 1.802 | 0.631 | 3.689 |
| $(1, 1, 1)$ | 7.76 | 0.511 | 2.050 | 0.761 | 5.256 |
| $(1, 3)$ | 7.18 | 0.527 | 2.063 | 0.793 | 5.408 |
| $(1, 1)$ | 7.14 | 0.576 | 2.280 | 0.868 | 6.195 |
| $(1)$ | 6.53 | 0.763 | 2.762 | 0.976 | 7.312 |

Table 4: Hyperparameter study

factors of 1, 3, and 9. As expected, increasing the number of layers in the C layers leads to improved results. This can be attributed to the fact that additional layers can capture more complex features, thereby enhancing the model's accuracy. To assess the impact of hierarchical dilated convolution, we compare the outcomes obtained with dilation rates $(1, 1)$ and $(1, 1, 1)$ against those acquired with dilation rates $(1, 3)$ and $(1, 3, 9)$. The results clearly demonstrate that hierarchical dilated convolution has a positive effect on the outcomes. This suggests that the ability to capture multiscale information through multiple dilation rates proves beneficial in enhancing the performance of the model.

## B.3 ABLATION STUDY

### B.3.1 ABLATION STUDY OF DCNO

| Model | Darcy rough | Trigonometric |
|---|---|---|
| DCNO(weighted loss) | 0.446 | 0.631 |
| DCNO($L^2$ loss) | 0.673 | 0.749 |
| DCNO(F layers only) | 0.973 | 1.554 |
| DCNO(C layers only) | 8.950 | 3.375 |
| DCNO(P kernel size of 3) | 0.537 | 0.631 |

Table 5: Ablation Study of DCNO.

In the ablation study presented in Table 5, our objective is to assess the impact of different components of the DCNO model on multiscale elliptic PDEs.

- By comparing with DCNO($L^2$ loss), we observe that utilizing the weighted loss function can lead to improved results. It is worth noting that even when employing the $L^2$ loss function, DCNO still outperforms other operator methods. We note that the remaining models in Table 5 also use the weighted loss.

- We further investigate the performance of DCNO(F layers only) and DCNO(C layers only) by removing specific components of the model. DCNO(F layers only), which retains only the F layers and can be seen as an enhanced version of FNO, does not achieve comparable accuracy to the full DCNO model. This indicates that the combination of F and C layers is crucial for achieving superior performance. On the other hand, DCNO(C layers only), which retains only the C layers and removes the F layers, exhibits significantly better performance in the multiscale trigonometric case compared to the Darcy rough case. This observation suggests that dilated convolutions in the C layers are particularly effective in improving results when dealing with rough coefficients. Comparing DCNO(F layers only) and DCNO(C layers only) highlights the importance of both the F layers and the C layers in the DCNO model. Both components play essential and complementary roles in achieving the superior performance demonstrated by the full DCNO model.

- To fully leverage the dataset in the examples of multiscale elliptic equations discussed in Section 4.1, the patch embedding function in DCNO employs a reduced stride. Specifically, when the resolution of the dataset divided by the resolution of the output is equal to $s$ ($s \geq 2$), a kernel size of 4 and a stride value of $s$ are used. This approach allows for effective utilization of the available data and addresses concerns about the use of additional input information to improve results. To further address the concern regarding the use of

additional input information, we conduct an experiment called DCNO (P kernel size of 3), which employs the same input information as other methods. We observe that incorporating more information improves the results for Darcy rough but has no impact on multiscale trigonometric coefficients. Importantly, even when using the same input information as other methods, DCNO consistently outperforms them, highlighting its superiority.

### B.3.2 ABLATION STUDY OF FNO

| Model | Darcy rough | Trigonometric |
|---|---|---|
| FNO(identity) | 1.925 | 1.918 |
| FNO(linear transform) | 1.749 | 1.803 |
| FNO(convolution) | 1.278 | 1.552 |

Table 6: Ablation Study of FNO.

In this study, we compare three choices for the second component of the F layers. We note that in the FNO framework introduced in Li et al. (2020b), a linear transformation denoted as $W$ is utilized. The three choices are as follows:

- FNO(identity): In this choice, the linear transformation $W$ is set to 1, resulting in an identical transformation.

- FNO(linear transform): This choice follows the implementation in Li et al. (2020b), where $W$ is a learned parameter, allowing for flexibility in the transformation.

- FNO(convolution): Instead of using a linear transformation, we employ a convolutional neural network (CNN) with a kernel size of 3 as a replacement for $W$.

By incorporating the CNN in the second component, FNO(convolution) demonstrates improved results by effectively capturing more fine-scale information. However, relying solely on a CNN in the second part is insufficient for accurately capturing high-frequency information, especially in multiscale problems where preserving high-resolution features is crucial. This limitation is evident in Table 5 of our paper. To address this challenge and effectively tackle multiscale problems, we employ a combination of F layers and C layers. This approach leverages the strengths of both layer types and enables better capturing of fine details at high resolutions, which is of significant importance in our study.

### B.3.3 COMPARISON OF DILATED CONVOLUTION AND U-NET

| Model | Parameters $\times 10^6$ | Memory (GB) | Time per epoch(s) | Darcy rough s=128 | s=256 | Trigonometric s=256 | s=512 |
|---|---|---|---|---|---|---|---|
| DCNO | 1.74 | 2.68 | 7.88 | 0.421 | 0.446 | 0.631 | 0.722 |
| U-NET+FNO | 4.89 | 11.51 | 21.73 | 0.390 | 0.441 | 0.500 | 0.504 |
| DCNO$^\star$ | 6.93 | 4.86 | 13.54 | 0.263 | 0.286 | 0.436 | 0.430 |

Table 7: Comparison of dilated convolution and U-Net.

U-Net is often considered a competitive alternative to dilated convolutions in the field of semantic segmentation (Ronneberger et al., 2015). To further investigate this, we conducted an experiment where we replaced the C layers in DCNO with a three-layer U-Net implementation, referred to as U-Net+FNO in Table 7. As shown in the table, U-Net+FNO achieved better results compared to the original DCNO, albeit at the cost of increased memory usage and training time due to the additional convolutions in the U-Net architecture. To ensure a fair comparison, we doubled the feature dimension of DCNO, resulting in a model referred to as DCNO$^\star$. Remarkably, DCNO$^\star$ exhibited superior accuracy compared to U-Net+FNO with fewer memory resources and faster training time. Taking these considerations into account, we conclude that dilated convolutions are better suited for operator learning tasks.

### B.4 STABILITY

In practical scenarios, it is common for the observed input data to be inaccurate and noisy. To assess the stability of neural operators in the forward problem, we introduced Gaussian noise, similar to that used in the inverse problem (refer to Section 4.3), to the input data. It is evident from the results presented in Table 8 and Figure B.1 that the predicted solutions generated by DCNO are minimally affected by the noise and remain highly accurate. These findings emphasize the robustness of neural operators when faced with noisy input data, making it a favorable alternative to traditional methods.

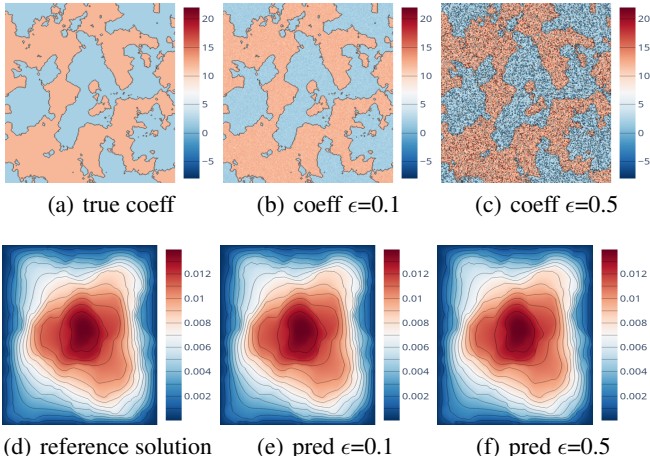

(a) true coeff     (b) coeff $\epsilon$=0.1     (c) coeff $\epsilon$=0.5

(d) reference solution     (e) pred $\epsilon$=0.1     (f) pred $\epsilon$=0.5

Figure B.1: DCNO solution for the Darcy rough forward problem with noise

| Model | Darcy rough | | | Trigonometric | | |
|---|---|---|---|---|---|---|
| | $\epsilon$=0.0 | $\epsilon$=0.1 | $\epsilon$=0.5 | $\epsilon$=0.0 | $\epsilon$=0.1 | $\epsilon$=0.5 |
| FNO | 1.749 | 1.750 | 1.806 | 1.803 | 1.887 | 2.109 |
| MWT | 1.301 | 1.320 | 1.448 | 0.988 | 1.061 | 1.249 |
| U-NO | 1.324 | 1.333 | 1.350 | 1.370 | 1.444 | 1.631 |
| GT | 2.166 | 2.203 | 2.274 | 1.013 | 1.038 | 1.422 |
| HANO | 1.119 | 1.201 | 1.211 | 0.743 | 1.033 | 1.159 |
| DIL-RESNET | 7.110 | 7.237 | 9.139 | 2.301 | 4.840 | 9.568 |
| DCNO | **0.446** | **0.450** | **0.524** | **0.631** | **0.642** | **0.833** |

Table 8: The stability of neural operators in the forward problem

## C SPECTRAL BIAS

The spectral bias, also known as the frequency principle, suggests that deep neural networks (DNNs) face challenges in effectively learning high-frequency components of functions that exhibit variations at multiple scales. This phenomenon has been extensively studied and discussed in the literature (Rahaman et al., 2019; Ronen et al., 2019; Xu et al., 2019) in the context of function approximation.

In the context of operator learning, we conducted an analysis on the relative error spectrum dynamics of a multiscale trigonometric example presented in Section 4.1. The analysis is illustrated in Figure C.1. To begin, we computed the Fourier transform of the relative error in the frequency domain $[-128\pi, 128\pi]^2$. Our examination focused on the error density $\rho(r)$ within the annulus $A(r)$, which satisfies the equation $\int_{A(r)} \rho(r) r dr = \sum_{i \in A(r)} \epsilon_i$. The annulus $A(r) := B(r+1) \backslash B(r)$, where $B(r)$ represents a sphere of radius $r$. The term $\epsilon_i$ represents the Fourier-transformed relative error at a lattice point indexed by $i$ in the discrete frequency domain. Notably, in the specific multiscale trigonometric example we examined, the energy of the solutions was concentrated within the domain $B(40\pi)$, as depicted in Figure D.1.

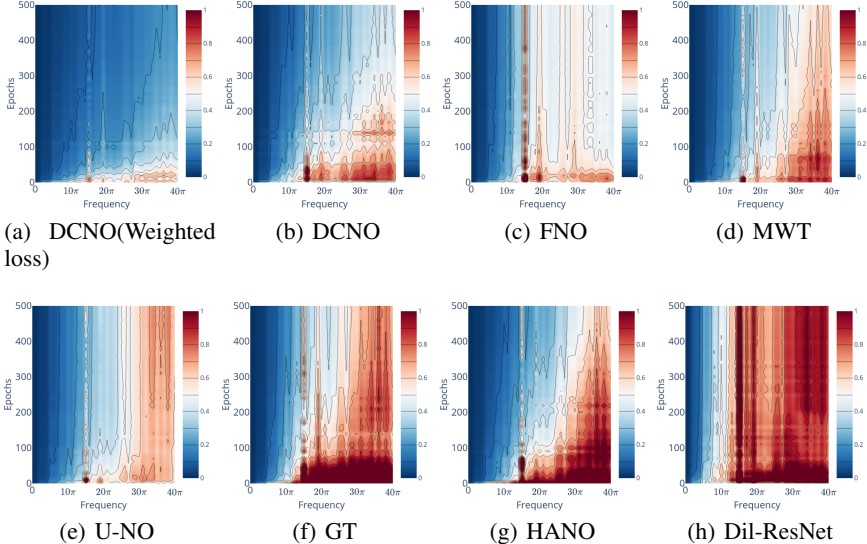

Figure C.1: Error dynamics in the frequency domain for multiscale trigonometric example.

In Figure C.1, the x-axis represents the first $40\pi$ dominant frequencies arranged from low frequency to high frequency, while the y-axis represents the number of training epochs. The plot reveals several significant findings. Firstly, the DCNO model demonstrated a faster decay of error for higher frequencies, indicating its ability to effectively capture high-frequency components. Additionally, the DCNO model maintained a more uniform reduction in errors across all frequencies, suggesting its proficiency in learning variations at multiple scales. Moreover, the DCNO model outperformed other methods, achieving lower testing errors and indicating its effectiveness in capturing and reducing errors across different frequency ranges. This analysis provides further evidence of the advantages of the DCNO model in effectively addressing the spectral bias in operator learning and accurately predicting functions with variations at multiple scales.

## D  WEIGHTED LOSS

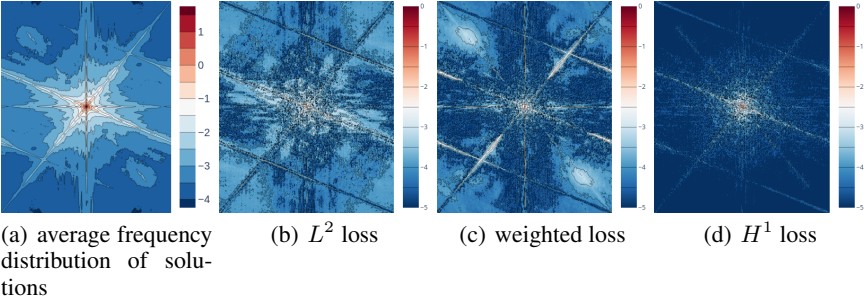

Figure D.1: (a)The frequency distribution of solutions associated with multiscale trigonometric coefficients (Appendix A.1.2); (b)(c)(d) absolute error spectrum of DCNO in $\log_{10}$ scale trained by $L^2$ loss, weighted loss and $H^1$ loss.

It is worth noting that the weighted loss function $L^T$ defined in Section 3.2 is equivalent to the $L^2$ loss when $T = 1$ and is equivalent to the $H^1$ loss when $T = N/2$. Figure D.1 (a) provides insight into the energy concentration of the solutions within the first 40 frequency modes. Subsequently, Figure D.1 (b), (c), and (d) display the absolute error spectrum of the DCNO model trained using the $L^2$ loss, weighted loss, and $H^1$ loss, respectively, represented on a logarithmic scale ($\log_{10}$).

| Model | Darcy rough | | Trigonometric | |
|---|---|---|---|---|
| | s=128 | s=256 | s=256 | s=512 |
| DCNO($L^2$ loss) | 0.587 | 0.673 | 0.749 | 0.836 |
| DCNO($H^1$ loss) | 0.583 | 0.680 | 0.785 | 0.897 |
| DCNO(weighted loss) | **0.421** | **0.446** | **0.631** | **0.722** |

Table 9: DCNO trained by different loss functions.

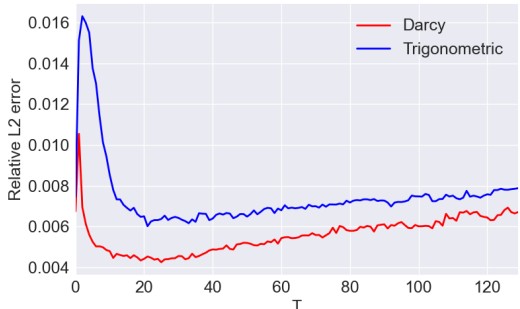

Figure D.2: Influence of $T$ for weighted loss function.

The figures clearly depict that training the DCNO model with the $H^1$ loss effectively reduces high-frequency errors by emphasizing high-frequency components through the inclusion of a term proportional to $|\xi|^2$ in the summation. However, upon reviewing Table 9, it becomes evident that there is no improvement in the results obtained with the $H^1$ loss compared to those achieved with the $L^2$ loss. This lack of improvement can be attributed to the relatively lower weight assigned to low-frequency components in the $H^1$ loss function.

Conversely, significant enhancements are observed when utilizing the weighted loss function, which assigns added weight to the lower modes. This weighting scheme effectively captures and reduces errors associated with these dominant low-frequency components, leading to improved performance compared to both the $L^2$ and $H^1$ loss functions.

To further investigate, Figure D.2 presents a comparative analysis of the results obtained by adding weight to the first $T$ modes in both the Darcy rough experiment and the multiscale trigonometric experiment. The figure clearly demonstrates that adding weight to the first $20 - 40$ modes for the multiscale trigonometric example and the first $15 - 30$ modes for the Darcy rough example leads to improved outcomes. These weighted modes effectively capture the essential components of the respective problems, resulting in enhanced performance. However, it is important to note that further increasing the number of weighted modes beyond a certain threshold leads to non-optimal results. This suggests that excessive emphasis on additional modes may introduce instability and disrupts the overall accuracy of the models.

# E    HELMHOLTZ EQUATIONS

We test the performance of DCNO for the acoustic Helmholtz equation in highly heterogeneous media as an example of multiscale wave phenomena, whose solution is considerably expensive for complicated and large geological models. We adapt the setup from Freese et al. (2021),

$$\begin{cases} -\mathrm{div}(a(x)\nabla u(x)) - \kappa^2 u = f(x), & x \in D, \\ u(x) = 0, & x \in \partial D. \end{cases}$$

where the coefficient $a(x)$ takes the value 1 or $\varepsilon$ as shown in Figure E with $\varepsilon^{-1} \in \mathrm{rand}(128, 256)$, $\kappa = 9$, and

$$f(x_1, x_2) = \begin{cases} 10^4 \exp\left( \dfrac{-1}{1 - \frac{(x_1 - 0.125)^2 + (x_2 - 0.5)^2}{0.05^2}} \right), & (x_1 - 0.125)^2 + (x_2 - 0.5)^2 < 0.05^2, \\ 0, & \text{else.} \end{cases}$$

| Model | Parameters $\times 10^6$ | Memory (GB) | Time per epoch(s) | Relative $L^2$ error s=128 | s=256 | s=512 |
|---|---|---|---|---|---|---|
| FNO | 2.37 | 1.79 | 5.61 | 5.033 | 5.405 | 6.295 |
| MWT | 9.81 | 2.54 | 17.74 | 2.653 | 2.731 | 2.875 |
| U-NO | 16.39 | 1.57 | 9.78 | 2.917 | 2.852 | 3.028 |
| GT | 2.22 | 9.32 | 35.36 | 13.505 | 13.828 | 16.601 |
| HANO | 13.37 | 9.87 | 27.53 | 2.757 | 2.806 | 3.067 |
| DCNO | 3.08 | 3.40 | 10.05 | **2.573** | **2.665** | **2.776** |

Table 10: Benchmarks on Helmholtz equations at various input resolution $s$. Performance are measured with relative $L^2$ errors ($\times 10^{-2}$), number of parameters, memory consumption for a batch size of 8, and time per epoch for $s = 256$ during the training process.

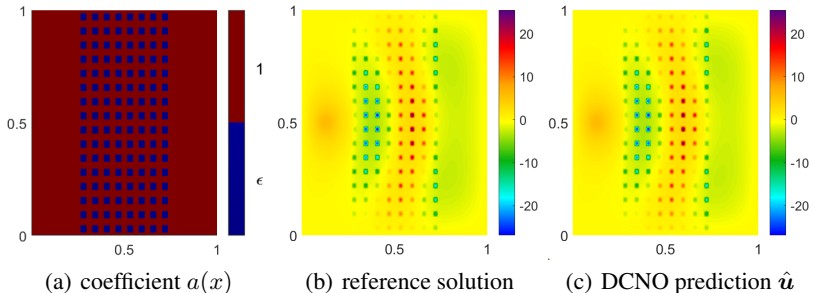

(a) coefficient $a(x)$     (b) reference solution     (c) DCNO prediction $\hat{u}$

Figure E.1: The mapping $a(x) \mapsto u$. (a) Heterogeneous coefficient $a(x)$, (b) the reference solution for $\varepsilon^{-1} = 237.3$, which is solved by $\mathcal{P}_1$ FEM implemented in FreeFEM++ (Hecht, 2012), (c)DCNO predicted solution

The Helmholtz equation poses a formidable challenge due to the highly oscillatory nature of its solution, as depicted in Figure E, as well as the presence of the $\kappa$-dependent pollution effect. However, the DCNO model has proven to be a remarkable solution, consistently surpassing other methods in effectively addressing this problem.

## F   ANOTHER NAVIER-STOKES EXAMPLE

| Model | Parameters $\times 10^6$ | Memory (GB) | Time per epoch(s) | relative $L^2$ loss $\times 10^{-2}$ |
|---|---|---|---|---|
| FNO | 2.37 | 0.52 | **5.71** | 0.117 |
| MWT | 9.81 | 0.64 | 33.04 | 0.087 |
| U-NO | 16.39 | **0.44** | 39.17 | 0.068 |
| GT | 2.22 | 2.33 | 23.43 | 1.296 |
| HANO | 13.37 | 2.58 | 22.35 | 0.078 |
| DIL-RESNET | **0.58** | 7.98 | 10.67 | 0.357 |
| DCNO | 3.05 | 0.82 | 7.42 | **0.062** |

Table 11: Benchmarks on incompressible Navier-Stokes equations. Performance are measured with relative $L^2$ errors ($\times 10^{-2}$), number of parameters, memory consumption for a batch size of 32, and time per epoch during the training process.

In this section, we investigate another Navier-Stokes example, as described in de Hoop et al. (2022). We continue to utilize the vorticity-stream function ($\omega - \psi$) formulation of the incompressible Navier-Stokes equations on a two-dimensional periodic domain denoted as $D = D_u = D_v = [0, 2\pi]^2$. Here, our objective is to learn the mapping from the forcing term $f$ to $v = \omega(\cdot, T)$, which represents the vorticity field at a given time $t = T$, i.e, $\mathcal{S} : f \mapsto \omega(\cdot, T) := \mathcal{S}(f)$. This formulation

differs from the previous example in terms of the governing equations and problem setup. The governing equations for this formulation are as follows:

$$\frac{\partial \omega}{\partial t} + (c \cdot \nabla)\omega - v\Delta\omega = f,$$

$$\omega = -\Delta\psi \quad \int_D \psi = 0,$$

$$c = \left(\frac{\partial\psi}{\partial x_2}, -\frac{\partial\psi}{\partial x_1}\right).$$

Table 11 presents the results of the experiments. The table reveals that the DCNO model achieves the lowest $L^2$ error among the compared methods, demonstrating its superior performance in terms of accuracy. Additionally, the DCNO model showcases favorable runtime, with only the FNO method surpassing it in terms of speed.

