# OpenReview forum: "Dilated convolution neural operator for multiscale partial differential equations"
_ICLR.cc/2024/Conference — Submitted to ICLR 2024_

### Official Review · Reviewer_TQWW · 2023-10-27

**Soundness:** 2 fair
**Presentation:** 2 fair
**Contribution:** 2 fair
**Rating:** 5
**Confidence:** 4

**Summary:**

This paper targets the solution of PDEs with a modified network architecture. The paper takes a neural operator approach, and builds on existing work, the popular fourier operators and dilated convolutional networks. The authors propose a specifc arrangement of fournier and conv blocks, which, as they claim, increases performance. In addition, a custom loss is introduced that requires a manual balancing of certain frequencies. The results are evaluated on two test sets, Darcy and NS, where the first one (the steady-state Darcy flows) is presented with variations, while the NS case is transient.

A variety of baselines, from relatively simple nets to more modern attentian based ones, are compared. Unfortunately, the networks seem to be from single training runs and use varying parameter counts, but given this state, the authors claim improvements in accuracy over the best performing baseline architectures.

**Strengths:**

The paper targets an important goal, namely the accurate and resource-aware prediction of PDE solutions. Handling this in a manner that takes into account multiple scales is an attractive goal, which, however, did not play a central role in the paper despite the title. Most modern architectures (apart maybe from very simple ResNets) inherently work on multiple scales, and the manual scale-separation and weighting in the loss seems to be problem specific.

It's also nice to see that the authors target an inverse problem, in addition to the regular "prediction" tasks.

**Weaknesses:**

Nonetheless, the datasets and tasks seem relatively simple, and are presented without much detail. E.g., no qualitative evaluations of the Darcy cases are presented, and its left unclear how the averaged MSE quantities influence the obtained solutions.

A central weakness that I see with this submission is the widely varying parameter count in the comparisons. It is well established that the size, i.e. parameter count, is one of the most influential factors determining NN performance. This is fine for most other networks, e.g., the U-NO network seems to have almost 10x parameters, and still shows a worse performance. Hence, I don't expect this to change much when a similar parameter count is used. It nonetheless would be a fairer comparison -- the large network could show some forms of overfitting. Most importantly, though, the dil-ResNets for some reason are given much fewer weights. I think it would be important to increase the number of features to obtain a parameter count on the level of the DCNO versions. They should do significantly better then, and it's not clear whether they would outperform the proposed approach.

A general question that is left open by the paper is that FNO and dil-ResNet by themselves do worse than the DCNO. Potentially, there could be an ingredient in the combined architecture that yields the "best of both worlds", but the evaluations in the paper leave this point open. If the authors could shed light on where the advantages come from, maybe with a careful ablation or additional analysis, I think this paper would be much more convincing. In combination with the relatively narrow range of experiments and sparse evaluation, I would be hesitant to try out the proposed architecture.

**Questions:**

- Most importantly, I am wondering how much is to be gained from the proposed architecture. Once the dil-ResNet is given the same number of parameters, and trained with the same loss, how does the performance compare to the proposed architecture?

- How many models were evaluated each time? Training deep nets is inherently stochastic, and the performance of the trained models varies. What are mean and standard deviation across multiple runs?

- As the approach targets multi-scale PDEs, and employs a scale separation in the proposed loss, have the authors evaluated frequency based evaluations and metrics?

- Where do the time ranges used for the different Reynolds numbers come from? The difficulty seems to "jump" up and down while increasing Re, and details are missing about how these tests were set up. Are these single test sequences, or were results averaged over multiple sequences to obtain stable results?

-----

I want to acknowledge the updates made by the authors and their answers to my questions. I think their work has potential, but for a fairly fundamental architectural advance it's still not fully convincing (judging from my concerns and those of the other reviewers). So I will keep my score and recommendation leaning towards the negative side, but encourage the authors to provide a more thorough "package" that makes clear where (and ideally why) the method works, and potentially also where it stops working.

---

> ### Author Response · Authors · 2023-11-23
> **Response to TQWW**
>
> We thank the reviewer for insightful comments and constructive suggestions.
>
> ### Response to Weaknesses
>
> In the revision, we have provided more detailed information about the experimental datasets and setup. Additionally, we have included an ablation study to analyze the impact of the parameter count on the model's performance.
>
> We are delighted that the reviewer acknowledged the concept of achieving the "best of both worlds". In the following, we will conduct a meticulous ablation study to address this point. To accomplish this, we propose integrating the two Fourier layers within the Decode part into the Process part of the architecture. This consolidation allows us to vary the number of Fourier (F) and convolutional ( C ) layers in the architecture and examine their impact on performance. By comparing results across various configurations, we can ascertain that DCNO represents the optimal combination of F and C layers.
>
> | F layers and C layers| Time per epoch(s)|Darcy rough($\times 10^{-2}$)|
> | ------------------------------- | ----------------- | ----------------------- |
> | 8 F layers                         |8.44 | 0.719 |
> | 7 F layers and 1 C layers(FCFFFFFF)|8.22 | 0.615 |
> | 6 F layers and 2 C layers(FCFCFFFF)|8.03 |0.563  |
> | 5 F layers and 3 C layers（DCNO)    |7.88 |**0.446** |
> | 4 F layers and 4 C layers (FCFCFCFC)      |7.66 |0.571 |
> | 3 F layers and 5 C layers (FCFCCCFC)      |7.47 |0.500 |
> | 2 F layers and 6 C layers (FCCCCCFC)      |7.27 |0.508 |
> | 1 F layers and 7 C layers (CCCCCCCF)     |6.96 |0.548  |
> | 8 C layers                      |6.52 |3.197 |
>
> It is important to note that we did not explore all possible combinations of F and C layers, and we adopted a notation like (FCFCFFFF) to denote the order of F and C layers.
>
> ### Response to Questions
>
> ### Question on Dil-ResNet with more parameters
>
> Please refer to "Experiment of Dil-Resnet with more parameters" in "Common Concerns".
>
> ### Question on the statistics of multiple runs
>
> We conducted all experiments using three different random seeds to ensure robustness and reliability of the results. We present the mean values and standard deviations of the reported results of DCNO($L^T$) in Table 1.
>
> | Model       | Darcy($L^2$ error  $\times 10^{-2}$) | Darcy($L^T$ error  $\times 10^{-2}$) | Trigonometric($L^2$ error $\times 10^{-2}$) | Trigonometric($L^T$ error$\times 10^{-2}$) |
> | ----------- | ---- | ---- | ------- | ------- |
> | DCNO($L^T$) | 0.446 $\pm0.013$  | 1.802$\pm0.054$                       |0.631$\pm0.017$          | 3.689$\pm0.127$          |
>
> ### Question on frequency based evaluations and metrics?
>
> In Table 1 of the revised paper, we conducted the Darcy rough and multiscale trigonometric experments using both $L^2$ loss and (frequency based) weighted $L^T$ loss. These experiments were also evaluated using two different error metrics: the $L^2$ error and the (frequency based) weighted $L^T$ error.
>
> ### Question on time ranges with increasing Reynolds numbers
>
> We refer to two papers [FNO](https://arxiv.org/abs/2010.08895) and [UNO](http://arxiv.org/abs/2204.11127), for the setup of time ranges with different Reynolds numbers adopted in this work. The general rule is to use a smaller simulation time for flows with higher Reynolds numbers.
>
> The observed variations in the results, which may appear as "jumping" up and down, could be attributed to different time ranges. We now perform an experiment using the same time range ($T_{in}=6, T=15$) for different Reynolds numbers:
>
> | Model | v=$1e^{-3}$            | v=$1e^{-4}$            | v=$1e^{-5}$            | v=$1e^{-6}$            |
> | ----- | --------- | ------ | -------- | ------- |
> | FNO  | $8.969 \times 10^{-4}$ | $9.116 \times 10^{-3}$ | $3.337 \times 10^{-2}$  | $5.280 \times 10^{-2}$ |
> | DCNO  | $8.151 \times 10^{-4}$ | $5.483 \times 10^{-3}$ | $1.707 \times 10^{-2}$ | $3.227 \times 10^{-2}$ |
>
> Based on the experiment results, it is consistent that the models exhibit higher accuracy for flows characterized by smaller Reynolds numbers.
>
> For the Navier-Stokes equations, the train-val-test split dataset has 5000, 500, and 500 samples. The results are averaged over 500 test datasets.

---

### Official Review · Reviewer_epq9 · 2023-10-31

**Soundness:** 3 good
**Presentation:** 3 good
**Contribution:** 2 fair
**Rating:** 5
**Confidence:** 4

**Summary:**

This work proposes a neural operator architecture (DNCO), that relies on the combination of fourier layers and dilated convolution layers. The former come from the original FNO architecture with some improvements, while the latter are inspired by works from *Holschneider et al.* and *Stachenfeld et al*. DNCO is compared to different baselines, mainly on the multiscale elliptic equations and the Navier-Stokes equations experiment from *Li et al*. Furthermore, experiments on the Helmholtz equations and a different case of the Navier-Stokes equations are contained in the appendix.

**Strengths:**

In my opinion, this work has three core strengths:

**S1:** Related work and similar methods are reviewed thoroughly, making it easy to contextualize this work, even for less experienced readers.

**S2:** The overall presentation is clear, easy to follow, and sufficient details are provided, especially in the methodology section.

**S3:** The paper considers a broad range of experiments across physical systems and different baseline architectures. The results are supplemented with additional ablations in the appendix.

In addition, source code is provided for this submission, which should help to improve the reproducibility of the shown results. However, I did neither investigate nor run the source code.

**Weaknesses:**

Apart from modifications to the FNO architecture, this paper has limited novelty and insights. While improvements of DCNO compared to the baselines appear across cases, it is not clear how well the parameters of each baseline are tuned. Especially so, as neither training details for the baselines and nor descriptions how key parameters were selected are included. In addition, I see the following problems:

### Formatting
The page formatting, page margins and/or the aspect ratio of this paper are clearly different from other submissions. This is especially noticeable due the amount of whitespace under the page number. But since most other formatting aspects are visible reasonable, I am not sure if this counts as a template/formatting violation.

### Evaluations
**E1:**
Why is only DCNO trained with the weighted loss, as it seems to be a highly important aspect in terms of performance (see Appendix B.3.1 and D). In my opinion, a fair evaluation would be to train all baseline methods with the weighted loss as well. Even though DCNO works better than the baselines using a simple L2 loss, this might not be the case for the weighted loss.

**E2:**
I think one of the core arguments of this paper, that DCNO performs better than previous methods in the high-frequency domain due to the dilated convolution layers, is not supported by the shown results. Fig. 1b only compares to FNO which is known to perform poorly on high frequencies. When considering other methods in Fig. A.1, DCNO only performs on par. To me, Fig. 1c is almost misleading, as the DCNO shown here was trained with the weighted loss that specifically focuses on high frequencies, while FNO in Fig. 1d was not. DCNO without this loss results in a significantly worse frequency behavior, as shown in Fig. C.1. Arguably, other baselines (especially MWT or HANO) even perform slightly better than the shown DCNO result in that figure, as they have smaller errors in the more crucial lower frequencies, indicated by the wider dark-blue area reaching over $10*\pi$. Another point regarding this issue is that a model with no dilations in the convolutions seems to result in a similar performance to DCNO (as shown in Tab. 4).

In my opinion, performing additional evaluations in this direction would be a highly necessary improvement. One option would be frequency analyses on the Navier-Stokes cases, similar to established techniques from that domain (see e.g. *“Turbulent flows”*, Pope, Cambridge University Press). Such evaluations would clearly show if DCNO exhibits fundamentally better high-frequency behavior compared to the other baselines.

**E3:**
Tab. 4 and Tab. 7 seem to show that models are underparameterized for the investigated learning tasks, as removing layers hurts performance and adding parameters substantially increases performance. To me, this indicates that other baselines such as dilated Resnets could perform better with a higher parameter count as well. As such, it would be nice to see how key parameters for the different baselines were selected, and if the performance could be improved when changing them.

**E4 (less crucial):**
If I interpret Appendix B.1 correctly, all results are computed on test sets that are randomly split from the training data. For PDE simulations, generalization to new parameters and initial conditions is a highly desirable property. Evaluations applying trained models to new parameter ranges, for example different Reynolds numbers in the fluid flow cases would be interesting to see.

**E5 (less crucial):**
As the data samples in the Navier-Stokes case are downsampled from the generation resolution of $256 \times 256$ to $64 \times 64$, does this not average out the effect of using higher Reynolds numbers? To that end it would also be interesting to see visualizations and corresponding predictions at different Reynolds numbers from this data set.

### Presentation:
There are some minor presentation issues listed below, and the structure of the appendix could be improved.

**Minor issues:**
- “allows us to selectively skip” → allows for selectively skipping (3. paragraph of Introduction)
- “convoluted” → convolved (several times in C layers description)
- “which maps the vorticity from time 0 to T0 to the vorticity from time T0 to a later time T” (unclear, Section 4.2)
- “Each convolution layer includes three convolutional neural networks, each” (not very clear, C layers description)
- Several citations lack publication information, e.g. *Cai et al., 2019,* *Freese et al., 2021*, *Liu et al. 2023*
- Citation format is inconsistent, e.g. multiple citations from arxiv are cited differently (arXiv e-prints, ArXiv, CoRR, …)
- Fig. 4.1: add log scaling to colorbar
- Fig. 4.2: add log scaling to plot axis description
- Subplots are plotted inconsistently across figures

### Summary
Overall, the weaknesses of this paper in its current state outweigh its strengths in my opinion. While the presentation of the paper is good, and it contains a broad range of experiments and results, the novelty/originality is limited and there are concerns regarding the evaluations. This leads to my overall recommendation of reject for this paper.

### Update after Author Response
While the author response addresses the presentation issues and the additional experiments close some of the gaps in terms of evaluations, I am still not fully convinced by the novelty and the overall usefulness of DCNO. Especially, I still have doubts with regard to the improved spectral behavior going beyond problem-specific adaptions.

Furthermore, it would be highly interesting to see if the observed improvements also hold for more complex learning tasks with more high-frequency content. With the additional visualizations and the supplementary material, it became visible that the investigated cases are quite smooth and not too complex. Even for the flow with a fairly low viscosity shown in Figure 4.3, most of the trajectory is very smooth (until about T=16s, when evaluating up to T=20s), and as such I think the contribution of the high-frequency information is minor across the aggregated loss evaluations.

Nevertheless, this work is investigating an interesting research direction, and the performance of DCNO on the shown problems are promising. As a result, I updated my review with the following changes:
- Soundness Score: increased from *2 fair* to *3 good*
- Overall Score: increased from *3 reject* to *5 marginally below the acceptance threshold*

**Questions:**

**Q1:**
Intuitively, dilated convolutions capture non-local, long-range, low frequency components due to the large receptive field that skips directly neighboring values. However, here the authors argue the opposite, which is quite counter-intuitive to me. Is there an intuition to this argumentation? Especially so, as removing dilations seems to result in very similar performance for the DCNO model (see Tab. 4), and the majority of the benefits in the frequency domain appear to come from the weighted loss (see Fig. C.1 as discussed above).

**Q2:**
Why does DCNO(F layers only) in Tab. 5 perform so much better than FNO (from Tab. 1)? Is this just due to better parameter tuning of the FNO layers?

**Q3:**
What is $s=128$, $s=256$, and $s=512$ in Tab. 1? I assume it might be the data resolution, but I did not explicit find a discussion of this aspect.

---

> ### Author Response · Authors · 2023-11-23
> **Response to epq9 (1)**
>
> We express our gratitude to the reviewer for the meticulous reading and insightful questions, as they have significantly contributed to improving the quality of our paper. In response to the reviewer's feedback, we have provided more details of the baselines and include more ablation studies to further enhance the clarity of our work.
>
> ### Response to Weakness
>
> We acknowledge that we may not have sufficiently emphasized the novelty of our method. The observed improvement in accuracy can be attributed to an optimal combination of modules that excel at extracting either low-frequency or high-frequency features. By effectively integrating these modules, our approach achieves superior performance compared to existing methods. Please also refer to the responses in "Common Concerns" and to the responses to the following specific questions.
>
>
> ### Formatting
>
> We are very greatful to the reviewer for pointing out the formatting issue, we have made corrections to address those issues.
>
> ### Evaluation
>
> ### E1
>
> In the revision, we performed training for all baseline methods using both the $L^2$ loss and the weighted $L^T$ loss, and evaluated these models using the relative $L^2$ error and the weighted $L^T$ error. The updated Table 1 provides compelling evidence that DCNO consistently maintains a substantial advantage over the other models, regardless of whether the $L^2$ loss or weighted loss is employed. While the weighted loss can contribute to improved accuracy, it is noteworthy that the DCNO model itself already demonstrates enhanced accuracy. In fact, DCNO trained with the $L^2$ loss outperforms most models trained with the weighted loss. Only HANO achieves comparable accuracy, albeit at the expense of three times longer training time.
>
> ### E2
>
> Thanks for the suggestion. In the revised paper, we have made a fair comparison by evaluating all models using the $L^2$ loss, as shown in Figure 1.1 and Figure A.1. The results clearly demonstrate that DCNO consistently achieves the highest accuracy among all the models, with or without the weighted loss.
>
> Furthermore, we agree that it is a little difficult to compare different models from Figure C.1. To provide a clearer representation of the loss dynamics, we have included Figure 1.1 (c,d) in the revision. In these figures, we decompose the error into low-frequency ($\leq 10\pi$, Figure 1.1.c) and high-frequency ($\geq 10\pi$, Figure 1.1.d) components. It is evident that DCNO surpasses most models (only HANO exhibits similair performance but with a training time more than 3 times longer than DCNO) in both the low-frequency and high-frequency components, further highlighting its superior performance.
>
> We have made updates to Table 4 in Appendix B.2. According to the revised table, for the Darcy rough case, a model with no dilation, denoted as (1) exhibits an $L^2$ error of 7.63%. With two layers (1,3), the error reduces to 5.27%, and with three layers (1,3,9), the error further decreases to 4.46%. This implies that the (1,3,9) configuration reduces the error by over 40%. Additionally, we not only observe an improvement with increasing layers but also note that the (1,1,1) setup has an $L^2$ error of 0.511, which is 15% higher compared to the (1,3,9) setup. This difference accounts for the improvement resulting from increasing dilation rates.
>
> We note that we carry out those studies for the Darcy case as Navier-Stokes cases require more time to train. In-depth investigation of the Navier-Stokes case will be conducted in future studies.
>
> ### E3
>
> Please refer to "Common Concerns" for the "Experiment of Dil-ResNet with more parameters".
>
> ### E4
>
> We train the model on the Navier-Stokes equation and test the model under different Reynolds numbers with $T_{in}=6, T=15$ .
> | DCNO ($T_{in}=6, T=15$) | test v=$1e^{-3}$ | test v=$1e^{-4}$ | test v=$1e^{-5}$ | test v=$1e^{-6}$ |
> | ----------------------- | ---------------- | ---------------- | ---------------- | ---------------- |
> | train v=$1e^{-3}$       |  | 0.3531   | 0.4038    | 0.4115    |
> | train v=$1e^{-4}$       | 0.4056    |      | 0.0923      | 0.1123   |
> | train v=$1e^{-5}$       | 0.4564   | 0.0885      |      |  0.0402     |
> | train v=$1e^{-6}$       | 0.4554       | 0.1031      | 0.0262 |       |
>
> It is evident that, in general, models trained with viscosities closer to the test cases exhibit reasonably good performances. An interesting observation is that all models trained with smaller viscosities, corresponding to higher Reynolds numbers, do not perform well for the $v=1e^{-3}$ case.
>
> ### E5
>
> In the revision, we have included a comparison between the Navier-Stokes (NS) solutions and DCNO predictions of the vorticity field, with viscosity $\nu=10^{-5}$, in Figure 4.3.
>
> Additionally, we add videos for the NS solutions and DCNO predicitons at different Reynolds numbers in the supplementary materials.

---

> ### Author Response · Authors · 2023-11-23
> **Response to epq9 (2)**
>
> ### Presentation
>
> Thanks for pointing out those presentation issues and we have addressed them in the revised paper.
>
> ### Response to Questions
>
> ### Q1 on dilated convolution
>
> Dilated convolution has the ability to capture relatively non-local and longer-range features by utilizing larger receptive fields. However, it falls short of achieving global context, such as the Fourier modes employed in the FNO model. Interestingly, the receptive fields of dilated convolutions bear resemblance to the coarse patches utilized in numerical homogenization methods [Malqvist & Peterseim 2014](https://www.ams.org/journals/mcom/2014-83-290/S0025-5718-2014-02868-8), [Hauck & Peterseim 2023](https://www.ams.org/journals/mcom/2023-92-341/S0025-5718-2022-03798-4/). Furthermore, through iterations across layers, information can propagate across receptive fields, akin to the domain decomposition analysis employed in the aforementioned numerical homogenization methods. Rigorous proofs using those techniques demonstrate that, for linear multiscale elliptic problems, it is possible to construct local bases over coarse patches with accuracy corresponding to the coarse patch size. To further improve the accuracy to fine grid size, we may need to add global bases [Benner et.al. 2018](https://epubs.siam.org/doi/10.1137/16M1098930). This insight leads us to the intuition that we can decompose the multiscale solution into low-rank global bases (e.g., Fourier modes) and local bases over coarse patches.
>
> The proposed DCNO architecture can be viewed as a neural network implementation of the iterative global-(quasi)local decomposition. This aspect contributes to its enhanced accuracy in multiscale operator learning.
>
> ### Q2 on FNO
>
> The reason DCNO(F layers only) in Tab. 5 perform better than FNO (from Tab. 1) is not due to parameter tuning of FNO layers. In fact, we have replaced the linear transformation in the second part of the Fourier layer with a 3x3 convolution. This has been explained in the description of F layers, and an ablation study can be found in Table 6 of Appendix B.3.2.
>
> To further study the effect of these two variants of FNO, we conduct experiments by increasing the number of fourier layers from 4 (FNO $L^T$) to 8 (FNO$\ast$($L^T$)) and trained them using the weighted $L^T$ loss. We also use 8 F layers as in DCNO, and denote it as FNO$\dagger$($L^T$).
>
> | Model            | Parameters($\times 10^6$) | Time per epoch(s) | Darcy($L^2$ error $\times 10^{-2}$) | Trigonometric($L^2$ error $\times 10^{-2}$) |
> | ---------------- | ------------------------- | ----------------- | ----------------------- | ------------------------------- |
> | FNO($L^T$) | 2.37 | 5.71   | 1.749    | 1.803 |
> | FNO$\ast$($L^T$) | 4.73| 10.19  |1.707 | 9.606 |
> | FNO$\dagger$($L^T$) |4.80  |10.64| 0.722 |0.962 |
> | DCNO$\ast$($L^T$) |1.71 | 7.55  |0.479 | 0.691 |
>
> It is observed that incorporating the 3x3 convolution in the modified FNO leads to improved accuracy. However, even with an additional 35% training time, the $L^2$ error of FNO$\dagger$ remains 50% higher than that of DCNO, for the Darcy Rough case.
>
> Indeed, considering the convolution layers already present in the DCNO architecture, we have come to the realization that the convolution operation in modified FNO may become redundant. This is evident from the last line of the aforementioned table, where we introduced the method DCNO*($L^T$). In this variant, we replaced the 3x3 convolution with the original FNO's pointwise linear transformation W. The results from the experiments clearly demonstrate that DCNO*($L^T$) with the F layer in the original FNO achieves comparable performance to DCNO with the modified F layer using a 3x3 convolution.
>
> ### Q3 on resolution $s$
>
> Yes, $s=128,256,512$ is the data resolution. We have added the description of $s$ in the caption of Table 1, and removed $s=128$ and $s=512$ cases.

---

### Official Review · Reviewer_Ds1k · 2023-10-31

**Soundness:** 2 fair
**Presentation:** 2 fair
**Contribution:** 1 poor
**Rating:** 5
**Confidence:** 4

**Summary:**

In this paper, the authors introduced DCNO (Dilated Convolution Neural Operator) as an effective approach for learning operators within multiscale partial differential equations (PDEs). The DCNO model adopted an Encode-Process-Decode architecture, starting with an encoder that employed a patch embedding function, which utilized a convolutional neural network (CNN) to elevate the input into a higher-dimensional feature space. The processor section alternated between Fourier layers (F layers) and Convolution layers (C layers), where F layers approximated low-frequency components, and C layers extracted high-frequency features. An ablation study was conducted to analyze the influence of these layers on model performance. Extensive experiments confirmed the effectiveness of DCNO in addressing multiscale PDEs, showcasing its superior performance and potential for various applications.

**Strengths:**

This practical applicability increases the significance of the research for industries and academic communities.

**Weaknesses:**

The framework seems to be an extension of a simple encoder-decoder architecture, with its modules combining F layers and C layers.

The contribution of this work lacks clarity, particularly in terms of the proposed decoder, which bears similarities to Cao (2021). It would be valuable for the authors to explicitly articulate the distinctions between these two approaches.

It is worth noting that dilated convolution is a well-established technology, and the novelty of the proposed Dilated Convolution Neural Operator (DCNO) seems somewhat limited. Authors are encouraged to provide a more robust justification for the uniqueness and innovation of their approach.

**Questions:**

The paper mentions DCNO as a novel architecture, but further clarification is needed to establish its novelty. The authors should explicitly detail how DCNO differs from existing approaches and what specific innovation it brings to the field.

The authors briefly mention similarities between their proposed decoder and a prior work by Cao (2021). It's crucial to provide a comprehensive comparison, highlighting the key differences and improvements in DCNO, to distinguish it from existing methods.

The absence of C layers in the decoder is notable, and the authors should explain this design choice. Providing a clear rationale for this decision would enhance the understanding of the model's architecture.

The inclusion of three convolutional neural networks in the convolution layer should be justified. The authors should elaborate on the reasoning behind this choice and discuss its impact on the model's performance and efficiency.

---

> ### Author Response · Authors · 2023-11-23
> **Response to Ds1k  (1)**
>
> We sincerely thank the reviewer for the valuable comments and constructive suggestions.
>
> ### Response to Weakness
>
> Firstly, we acknowledage that the Encode-Process-Decode framework is simple and widely applicable, and the dilated convolution is a well-established technology. However, the key factor behind DCNO's improved accuracy for multiscale PDEs lies in the optimal combination of models that are specifically designed to excel at extracting both local and global features, inspired by the global-local decomposition of multiscale solution space. Please also refer to the reponse on "Novelty and Contribution" in "Common Concerns".
>
> When comparing our framework to transformer-based methods, specifically the approach proposed by Cao (2021), it is important to acknowledge that our focus lies not on the Decode and Encode parts, which we share in common. Rather, the notable distinction between our approach and Cao (2021) lies in the Process part. While Cao (2021) relies on transformer/attention models, we take a different path by employing convolution/Fourier models. This clear methodological divergence accounts for the differences in their respective performances.
>
> ### Question on the novelty of DCNO and distinction with other models
>
> FNO type methods are well-suited for dealing with smooth solutions, but they often exhibit spectral bias when applied to multiscale problems. On the other hand, convolution-based methods such as Dil-Resnet and CNN offer fast and straightforward implementation and excel at extracting local features. However, they still struggle to effectively handle multiscale problems, as we see in the experiment results. Transformer-based methods like HANO (based on SWIN implementation) have been specifically designed to tackle multiscale operator learning tasks. Nonetheless, their implementation tends to be more complex, posing challenges in achieving an optimal cost-accuracy trade-off.
>
> Please see the "Common Concerns" for the novelty and contribution of DCNO. In fact, DCNO draws inspiration from recent advancements in multiscale computational methods, specifically the concept of representing the multiscale parameterized solution as the sum of low-rank global bases (analogous to low-frequency Fourier modes used in FNO) and localized bases on coarse patches (analogous to dilated convolution). This combination of techniques has proven instrumental in achieving outstanding results.
>
> ### Question on the decoder
>
> The primary distinction between DCNO and GT (Cao 2021) lies in the Process part. DCNO employs a hybrid architecture that encompasses Fourier layers and convolution layers, enabling the capture of both low-frequency and high-frequency information. In contrast, GT utilizes a Galerkin transformer to extract features.
>
> In our current implementation, the decoder of DCNO and GT is nearly identical. Both architectures consist of two Fourier layers in the decoder, while the second component of the Fourier layers in DCNO incorporates a convolution with a kernel size of 3, which replaces the local (pointwise) linear transform W utilized in Li et al. (2020b).
>
> We include an experiment where GT uses the DCNO decoder, and they are both trained by the weighted $L^T$ loss. We found that the performance (in $L^2$ error) of the two models is similar.
>
> | Model | Darcy rough ($\times 10^{-2}$) | Trigonometric($\times 10^{-2}$) |
> | ----- | ----------------------- | ------------------------------- |
> | GT|      1.739        |        0.988               |
> | GT (DCNO decoder)|     1.567    |    0.954|
>
>
> To further investigate the impact of the Decoder design, we conducted an ablation study by replacing one F layer with one C layer in the Decoder. We denote this variant as DCNO$\star$.
>
> | F layers and C layers | Time per epoch(s) | Darcy($\times 10^{-2}$) |
> | ------------------- | ----------------- | -----------------
> |  DCNO |7.88 |**0.446**         |
> |  DCNO$\star$       |7.66 |0.571             |
>
> The results suggest that the decoder architecture in DCNO outperforms DCNO$\star$.
>
> To improve notation clarity, we have made adjustments in the revision, such that all the F layers are consolidated into the Process part, while designating only the Feed-Forward Network (FFN) as the decoder.  In this way, we can flexibly adjust the number of Fourier (F) and convolutional (C) layers in the configuration. Through experiments, we have discovered that DCNO represents the optimal combination of F and C layers (we have not specifically tested the impact of different layer orders in our experiments). Please refer to our response to TQWW for further details.

---

> ### Author Response · Authors · 2023-11-23
> **Response to Ds1k  (2)**
>
> ### Question on the inclusion of three dilated convolutions Convolutional Neural Networks in the C layer
>
> The inclusion of three convolutional neural networks in the convolution layer is to capture the high-frequency information over larger receptive field. The impact of the number of layers and dilation rates is shown in the updated Table 4 of Appendix B.2.
>
> | Dilation  | Time per epoch(s) | Darcy rough($L^2 \times 10^{-2}$) | Darcy rough($L^T \times 10^{-2}$) | Trigonometric($L^2 \times 10^{-2}$) | Trigonometric($L^T \times 10^{-2}$) |
> | --------- | ------- | -------| -------- | ------- | -------- |
> | (1, 3, 9) | 7.88  | 0.446  | 1.802 | 0.631  | 3.689 |
> | (1, 1, 1) | 7.76  | 0.511       |2.050   | 0.761| 5.256 |
> | (1, 3)    | 7.18  | 0.527   |2.063 | 0.793   |5.408 |
> | (1, 1)    | 7.14  | 0.576 | 2.280  | 0.868  |6.195|
> | (1)       | 6.53 |0.763  | 2.762 | 0.976   |7.312 |
>
> We conducted a hyperparameter study to investigate the influence of different dilation rates in the C layers of the DCNO model for multiscale elliptic PDEs. The results of this study are trained with $L^2$ loss and summarized above. The dilation rates determine the configuration of the C layers in the DCNO model, where dilation rates of $(1, 3, 9)$ means that the C layers consist of three dilated convolutions with dilation factors of 1, 3, and 9.
>
> As expected, increasing the number of layers in the C layers leads to improved results. This can be attributed to the fact that additional layers can capture more complex features, thereby enhancing the model's accuracy. To assess the impact of hierarchical dilated convolution, we compare the outcomes obtained with dilation rates $(1, 1)$ and $(1, 1, 1)$ against those acquired with dilation rates $(1, 3)$ and $(1, 3, 9)$. The results clearly demonstrate that hierarchical dilated convolution has a positive effect on the outcomes. This suggests that the ability to capture multiscale information through multiple dilation rates proves beneficial in enhancing the performance of the model.

---

### Author Response · Authors · 2023-11-23
**Common Concerns**

We sincerely thank all reviewers for insightful comments and constructive suggestions. We will first address some common concerns.

### Novelty and Contribution

We would like to emphasize that the primary novelty and key contribution of our method lies in the integration of dilated convolution, which excels in extracting local features, and Fourier layers, which is effective at capturing global features. Notably, both components can be efficiently implemented on GPUs. This approach draws inspiration from recent advancements in multiscale computational methods, namely, the representation of multiscale parametric solutions as a combination of low-rank global bases and localized bases over coarse patches. This representation has demonstrated the ability to strike an optimal cost-accuracy trade-off. The local-global decomposition of multiscale solution spaces has been rigorously proved for linear multiscale PDEs. Our approach offers an alternative solution to hierarchical methods such as multilevel/multigrid approaches or hierarchical attention based neural operators, providing a novel and efficient approach to multiscale operator learning.


### Experiment of Dil-Resnet with more parameters

To address concerns regarding fairness in the experimental setups, we ensured that the settings for all baseline models were adopted exactly as presented in their respective papers, without any modifications. One crucial parameter that plays a significant role is the feature dimension, which corresponds to the width of the neural network.

In our experiments, we set the feature dimension to 32 for FNO, MWT, GT, and UNO, while Dil-Resnet utilized a feature dimension of 48. However, for the purpose of creating a lightweight model, we intentionally selected a smaller feature dimension of 24 for DCNO. In Table 7, we observed a noteworthy improvement in the results when we doubled the feature dimension of DCNO.

It is important to note that increasing the dimension of the feature space does not always guarantee improved results. To demonstrate this, we double the feature dimension of Dil-Resnet from 48 (DilResnet($L^T$)) to 96 (DilResnet*($L^T$)) and  triple the number of layers from 4 (DilResnet*($L^T$)) to 12 (DilResnet**($L^T$)), and trained them using the weighted $L^T$ loss.

| Model            | Parameters($\times 10^6$) | Time per epoch(s) | Darcy($L^2$ error $\times 10^{-2}$) | Trigonometric($L^2$ error $\times 10^{-2}$) |
| ---------------- | ------------------------- | ----------------- | ----------------------- | ------------------------------- |
| DilResnet($L^T$) | 0.58 | 10.69   | 5.202    | 1.848 |
| DilResnet*($L^T$) |2.33  |24.60| 5.689    | 1.720 |
| DilResnet**($L^T$) |6.98  |72.36| 2.002    |1.265  |

It is evident that despite having a significantly larger number of parameters ($6.98×10^6$, 4 times that of DCNO) and longer training time (72.36, 9 times that of DCNO), the error is 4.5 times higher compared to DCNO, for the Darcy rough case.

### Major revision to the paper

- We added the (testing) error dynamics for high-frequency ($> 10\pi$) and low- frequency ($\leq 10\pi$) components in Figure 1.1 ( c ) ( d )
- We added a paragraph in Section 2.1 (Numerical Methods for Multiscale PDEs) and a paragraph in Section 2.3 (Dilated convolution) to introduce the connection of dilated convolution with local bases in mulitscale numerical methods and the global-local decompostion of multiscale solutions.
- We updated Table 1, where all the models are trained with both the $L^2$ and the weighted loss ($L^T$), and the results are evaluted using both relative $L^2$ and $L^T$ errors.
- We added a figure (Figure 4.3) to visualize the results of Navier-Stokes equation.

---

### Meta-Review · Area_Chair_UxeF · 2023-12-05

**Metareview:**

Overall, the final ratings from the three reviewers tend to be negative. After a careful review of the reviewers' comments and the rebuttals provided by the authors, most reviewers acknowledge the potential of the research direction proposed in the work. However, there is a consensus among the three reviewers that there may still be shortcomings in the novelty of the work. Even after the authors' rebuttal, Reviewer epq9 and Reviewer TQWW maintain their previous views on the lack of novelty in the work. Furthermore, Reviewer epq9 emphasizes his concerns about novelty and the effectiveness of the method. In conclusion, while the potential value of the research direction is acknowledged, there may still be room for improvement in terms of novelty. Therefore, I believe the work has not met the standards for ICLR 2023, and I decide to reject it.

**Justification For Why Not Higher Score:**

Please refer to the metareview.

**Justification For Why Not Lower Score:**

N/A

---

### Decision · Program_Chairs · 2024-01-16

Reject